

# Geochemical signatures and nanomechanical properties of echinoid tests from nearshore habitats of Florida: environmental and physiological controls on echinoid biomineralization

Przemysław Gorzelak[1], Luis Torres Jr.[2], Dorota Kołbuk[3], Tobias B. Grun[4] and Michał Kowalewski[2]

[1] Institute of Paleobiology Polish Academy of Sciences, Warsaw, Poland
[2] Florida Museum of Natural History, University of Florida, Gainesville, FL, United States of America
[3] UCD Earth Institute and School of Biology and Environmental Science, Science Centre West, University College Dublin, Dublin, Ireland
[4] Department of Fundamentals of Nature Conservation and Data Management, Bavarian State Office for the Environment, Hof, Germany

Corresponding author
Przemysław Gorzelak,
pgorzelak@twarda.pan.pl

## ABSTRACT

The mechanisms that regulate minor and trace element biomineralization in the echinoid skeleton can be primarily controlled biologically (*i.e.*, by the organism and its vital effects) or by extrinsic environmental factors. Assessing the relative role of those controls is essential for understanding echinoid biomineralization, taphonomy, diagenesis, and their potential as geochemical archives. In this study, we (1) contrast geochemical signatures of specimens collected across multiple taxa and environmental settings to assess *in situ* the effects of environmental and physiological factors on skeletal biomineralogy; and (2) analyze the nanomechanical properties of the echinoid skeleton to assess potential linkages between magnesium/calcium (Mg/Ca) ratios and skeletal nanohardness. Live specimens of sand dollars and sea biscuits (*Mellita tenuis*, *Encope* spp., *Leodia sexiesperforata*, and *Clypeaster subdepressus*) were collected from three different salinity regimes: (1) a coastal region of Cedar Key influenced by freshwater input from Suwannee River, with low and fluctuating salinity; (2) St. James Bay with less fluctuating, higher salinity; and (3) Florida Keys with stable, fully marine salinity conditions. No clear relationship was found between the bulk skeletal barium/calcium (Ba/Ca), zinc/calcium (Zn/Ca), sodium/calcium (Na/Ca), cadmium/calcium (Cd/Ca), copper/calcium (Cu/Ca), phosphorous/calcium (P/Ca), lead/calcium (Pb/Ca), boron/calcium (B/Ca), manganese/calcium (Mn/Ca) ratios pooled across all taxa. In contrast, bulk Mg/Ca, strontium/calcium (Sr/Ca), sulfur/calcium (S/Ca) and lithium/calcium (Li/Ca) ratios exhibited notable differences between the three regions, indicating that distribution of these elements can be at least partly influenced by environmental factors such as salinity. However, such patterns were highly variable across taxa and regions, indicating that both environmental and physiological factors influenced geochemical signatures to varying degrees, depending on the species and environmental setting. In addition, regardless of species identity, different types of stereom within single tests were characterized by distinct skeletal Mg/Ca ratios and nanohardness. The inner galleried and coarse labyrinthic stereom typically exhibited
a lower Mg/Ca ratio and nanohardness than the outer imperforate stereom layer that locally forms tubercles. Such heterogeneity in Mg distribution within single specimens cannot be ascribed solely to environmental changes, indicating that these echinoids actively regulate their intraskeletal Mg content: the higher magnesium concentration at the tubercles relative to that of the underlying stereom may be interpreted as a strategy for enhancing their mechanical strength to withstand surface friction and wear. The results suggest that the trace element composition of echinoid tests is a complex outcome of environmental and physiological factors.

## INTRODUCTION

Echinoderms produce magnesium (Mg) calcite skeletons with a unique trabecular microstructure (stereom) of mesodermal origin. There is growing evidence that the ultimate skeletal chemistry in echinoderms is a result of a complex interplay between environmental and physiological factors (*e.g.*, *Weber, 1969*; *Weber, 1973*; *Solovjev, 2014*; *Smith et al., 2016*; *Iglikowska et al., 2020*; *Azcárate-García, Avila & Figuerola, 2024*). Yet the extent to which these factors control the incorporation of minor and trace elements into the echinoderm skeleton is poorly known. Whereas a number of insightful experimental studies under controlled laboratory conditions have been performed (*e.g.*, *Ries, 2004*; *Borremans et al., 2009*; *Hermans et al., 2010*; *Asnaghi et al., 2014*; *Duquette et al., 2018*; *Kołbuk et al., 2019*; *Kołbuk et al., 2020*; *Kołbuk et al., 2021*), interpretations of such experimentally collected data are nontrivial due to potential pitfalls related to the short time-scale of experiments (*e.g.*, *Knapp et al., 2012*), constraints on other factors that may influence biomineralization in natural systems, and "shock" effects of the treatments. Conversely, contrasting specimens collected from the environments they inhabited provide natural experimental settings for assessing the effect of environmental parameters on skeletal chemistry *in situ*. However, in natural settings environmental parameters often covary and species assemblies vary in taxonomic composition, hampering research design and hypothesis testing.

Most previous studies, both experimental and those based on field-collected specimens, primarily focused on the concentrations of major elements within the calcite skeleton of echinoderms, namely magnesium (*e.g.*, *Chave, 1954*; *Weber, 1969*; *Weber, 1973*; *Richter & Bruckschen, 1998*; *Smith et al., 2016*; *Kołbuk et al., 2020*; *Azcárate-García, Avila & Figuerola, 2024*). Overall, these studies have demonstrated that the skeletal magnesium/calcium (Mg/Ca) ratio in echinoderms can be positively correlated to environmental parameters, such as ambient Mg/Ca sea water ratio, temperature, and salinity. However, echinoderms from the same locations may display a wide range of skeletal Mg contents (*e.g.*, *Chave, 1954*; *Weber, 1969*; *Gorzelak et al., 2013*; *Iglikowska et al., 2017*; *Iglikowska et al., 2018*), which may vary at hierarchical scales (within a single stereom trabecula, within ossicle, among ossicles within a single individual, and at higher taxonomic levels). Therefore, genetic factors are also important in modulating magnesium distribution. Notably, Mg/Ca
ratios in echinoderm skeletons can be affected by the type of diet (*Asnaghi et al., 2014*; *Kołbuk et al., 2020*), as well as the type and quantity of the intra-stereomic organic matrix involved in biomineralization (*Hermans et al., 2011*; *Gorzelak et al., 2013*).

Apart from magnesium, the echinoderm skeleton contains various admixtures of minor and trace elements, but only a few studies have dealt with their analyses (*Lebrato et al., 2013*) and/or their possible connections with environmental parameters. For instance, in the case of strontium, it has been experimentally shown that the skeletal strontium/calcium (Sr/Ca) ratio in asteroids can be positively linked to temperature and salinity (*Borremans et al., 2009*). Conversely, the species-specific trace element composition in Arctic echinoderms was interpreted as indicative of biological control, though accumulation of some metals and minor elements was primarily shaped by ambient conditions (*Iglikowska et al., 2020*).

Here, we report the bulk ratios of selected minor and trace elements in the skeletons of clypeasteroids (Clypeasteroidea) and scutelloids (Scutelloida), commonly known as sand dollars and sea biscuits, which were collected from various salinity regimes throughout Florida in order to assess the extent to which these ratios are environmentally controlled, with particular focus on salinity. It is assumed here that significant differences in elemental ratios among environmental settings, especially within a given species, may be indicative of the influence of extrinsic factors. Conversely, significant differences in geochemical signatures between sympatric species or comparable signatures for conspecific specimens from different environmental settings may be indicative of physiological controls.

Sand dollars are flat, shallow burrowing sea urchins that have been extensively studied in the context of their ecology, taphonomy and skeletal morphology (*e.g.*, *Li et al., 2013*; *Guilherme, Brustolin & Bueno, 2015*; *Brustolin et al., 2016*; *Grun & Nebelsick, 2016*; *Grun & Nebelsick, 2018*; *Mancosu & Nebelsick, 2017*; *Nebelsick, 2020*; *Lin et al., 2021*; *Cleveland & Pomory, 2022*). However, relatively little attention has been given to their skeletal chemistry (*Clarke & Wheeler, 1922*; *Chave, 1954*; *Weber, 1969*; *Macqueen, Ghent & Davies, 1974*; *Solovjev, 2014*; *Smith et al., 2016*; *Perricone et al., 2021*). This paper aimed at filling this gap. Our data are supplemented by micro-scale spot elemental analyses aimed at assessing the possible intraskeletal variation in Mg/Ca ratio, and its relation to nanomechanical properties (nanohardness) of the stereom. This allows us to verify data obtained from experiments (*e.g.*, *Gorzelak et al., 2024*) suggesting that impurities of some elements (such as Mg) may affect mechanical properties of echinoid biocalcite, which may be of functional and adaptive significance.

## MATERIALS & METHODS

### Study systems and data collection

Live echinoid specimens were collected *via* SCUBA expeditions from 15 sites (Table 1) in three regions along the western and southern coasts of Florida: (1) central Florida Keys (FK), (2) Cedar Key (CK), and (3) Apalachee Bay at the northern Gulf coast of Florida (NG) (Fig. 1). All surveying and collecting activities were carried out within the scope of the collecting permits SAL-18-1294A-SR, SAL-19-2195-SR, SAL-19-2195A-SR, SAL-22-2195-SR, and SAL-22-2195A-SR issued by the Florida Fish and Wildlife Conservation

**Table 1  Summary of sampling sites.**

| Region | Site | Latitude | Longitude | Depth [m] | *Clypeaster subdepressus* | *Encope* spp.[*] | *Leodia sexiesperforata* | *Mellita tenuis* |
|---|---|---|---|---|---|---|---|---|
| *by site* | | | | | | | | |
| Northern Gulf[**] | CK-2 | 28.99317 | −83.1371 | 7.6 | 1 | 0 | 0 | 1 |
| Cedar Key[**] | CK-5 | 28.9946 | −83.1424 | 4 | 0 | 0 | 0 | 3 |
| Cedar Key | CK-6 | 29.06557 | −83.0656 | 0.9 | 0 | 0 | 0 | 3 |
| Florida Keys | FK-17 | 24.81612 | −80.7353 | 6.1 | 0 | 0 | 4 | 0 |
| Florida Keys | FK-3 | 24.78605 | −80.8829 | 1.8 | 0 | 0 | 3 | 0 |
| Florida Keys | FK-5 | 24.74398 | −80.7988 | 8.5 | 1 | 0 | 0 | 0 |
| Florida Keys | FK-9 | 24.74689 | −80.7945 | 8.2 | 0 | 2 | 2 | 0 |
| Northern Gulf | AP-10 | 29.76553 | −84.5447 | 12.2 | 0 | 3 | 0 | 0 |
| Northern Gulf | AP-15 | 29.61218 | −84.3702 | 20.7 | 0 | 3 | 0 | 0 |
| Northern Gulf | AP-17 | 29.83326 | −84.5276 | 7.6 | 0 | 0 | 0 | 5 |
| Northern Gulf | AP-19 | 29.73817 | −84.2415 | 14 | 1 | 0 | 0 | 0 |
| Northern Gulf | AP-8 | 29.8427 | −84.2712 | 11 | 0 | 1 | 0 | 0 |
| Northern Gulf | CB-4 | 29.8042 | −84.4941 | 7 | 2 | 0 | 5 | 0 |
| Northern Gulf | CB-5 | 29.81896 | −84.4758 | 7.3 | 0 | 0 | 0 | 1 |
| Northern Gulf | ST-1 | 29.50526 | −83.6723 | 11.3 | 0 | 2 | 0 | 0 |
| *by region* | | | | | | | | |
| Cedar Key[**] | 2 sites | N/A | N/A | N/A | 0 | 0 | 0 | 6 |
| Florida Keys | 4 sites | N/A | N/A | N/A | 1 | 2 | 9 | 0 |
| Northern Gulf | 9 sites | N/A | N/A | N/A | 4 | 9 | 5 | 7 |
| | | Total | ($n = 43$) | | 5 | 11 | 14 | 13 |

**Notes.**

[*] *Encope* spp. include sister taxa *Encope michelini* and *Encope aberrans*.

[**] CK-2 is an offshore site off Cedar Key located outside the direct Suwannee River influence and is classified as "Northern Gulf" with salinity estimates based on a comparable site located southward of Cedar Key (see text for more details).

Relative to CK-2 and CK-6, CK-5 is an intermediate site shallower and more proximal to the estuary than CK-2, but more offshore and deeper than CK-6. It was classified in the main analysis as "Northern Gulf" but was reclassified as "Cedar Key" and assigned "Cedar Key" salinity in the supplementary analyses. The results of alternative salinity/region assignments influence minor interpretative and statistical details, but do not alter any of the major conclusions of the study.

Commission. A total of 43 specimens were analyzed for bulk geochemical analysis and four of those specimens were also subject to nanoindentation and fine scale microprobe analyses.

The three regions notably differ in salinity and, to a smaller extent, in temperature. FK is characterized by the highest and least variable salinity and temperature, NG is characterized by relatively lower and more variable salinity and temperature, and CK sites, influenced by Suwannee estuary, have the lowest salinity (Fig. 2) and temperature comparable to FK. The FK sites represent carbonate sand bottoms, whereas CK and NG sites represent mixed siliciclastic-carbonate sand bottoms. All sites occur in shallow-water, coastal areas with water depth ranging from 1 to 21 m (Table 1). The offshore site off Cedar Key (site CK-2) is outside the direct influence of Suwannee river and thus more comparable to other Northern Gulf sites. Salinity estimates for CK-2 were obtained from a comparably offshore, non-estuarine site off Crystal River with salinity estimates consistent with an *in situ* measurement of 31‰ collected at CK-2 during field work. In all analyses, site CK-2 is classified as 'Northern Gulf' (Table 1). The shallower offshore site (CK-5) was classified as

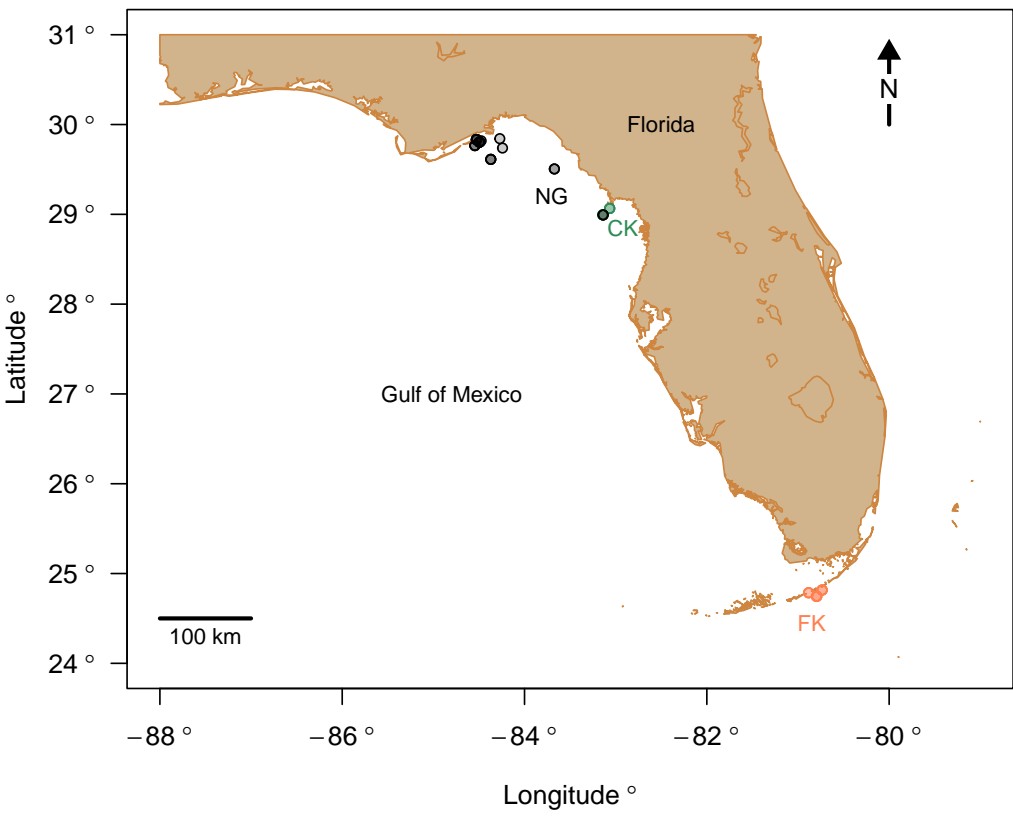

**Figure 1** **Study area map with the sampling sites color-coded by region.** Abbreviations: FK, Florida Keys; CK, Cedar Key; NG, Northern Gulf. Map plotted in R (*R Core Team, 2023*) using publicly available NOAA coordinates downloaded from https://gnome.orr.noaa.gov/goods/tools/GSHHS/coast_subset. Points marking sampling sites are color-coded by region: green, Cedar Key sites; orange, Florida Key sites; gray, Northern Gulf sites. The site symbols are semi-transparent and darker shade indicate multiple sites plotting at the same position at the coarse spatial resolution of the map.

'Cedar Key' with salinity the same as CK-6. The alternative results, with CK-5 assigned to "Northern Gulf" using the same salinity estimates as for the nearby CK-2, are also reported below. In addition, the CK-5 specimens are highlighted with special symbols on the key figures included below. Reassigning regional membership and salinity for sites CK-5 and CK-2 changes slightly some of the plots and values of statistics reported below, but does not change any of the major conclusions of this study. Reassignments of CK-5 and CK-2 to different regions and salinity estimates were performed interactively during the analysis (see R script in Data S1), whereas the values archived in Data S2 are original values prior to the reassignment.

Four irregular echinoid taxa were selected based on their common presence in the studied regions and adequately large test size. All taxa included here possess an endoskeleton (test) consisting of interlocking plates made of high-Mg calcite.

*Clypeaster subdepressus* (family Clypeasteridae) is a relatively large echinoid common in soft-bottom sandy habitats of the Gulf of Mexico. The species is a mobile, semi-infaunal to

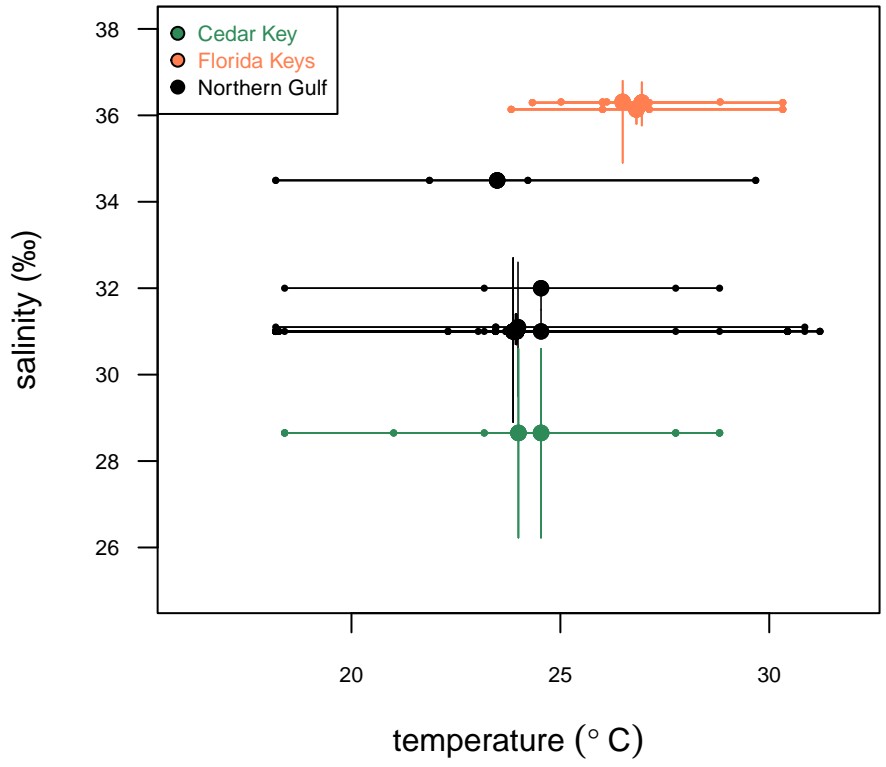

**Figure 2** **Comparison of the three study systems (FK, Florida Keys; CK, Cedar Keys, and NG, Northern Gulf) in terms of temperature and salinity.** Variation in surface temperature (horizontal axis) across the study systems. Large dots represent mean annual temperature values, small dots represent seasonal (winter, spring, summer, fall) estimates, and bars represent seasonal temperature range. Variation in salinity (vertical axis) across the three study systems. Large dots represent mean annual salinity values and bars represent interquartile salinity ranges. Temperature and salinity data extracted from *NOAA (2024)* and *Gulf of Mexico Ocean Observation (GCOOS)(2019)* are provided in Data S2. Symbols and bars color-coded by region: green, Cedar Key sites; orange, Florida Key sites; gray, Northern Gulf sites.

epifaunal detritivore. During collecting, the specimens were typically observed only partly buried under sediment surface with the aboral side protruding above sediment surface (see also *Telford, Mooi & Harold, 1987*). *C. subdepressus* ingests coarse sand-sized particles and mostly feeds on dead algae, seagrass, and coral fragments (*Telford, Mooi & Harold, 1987*).

*Encope* spp. include two species of sand dollars: *E. michelini* and *E. aberrans* (family Mellitidae). Juveniles and subadults of these two species are difficult to distinguish morphologically and the molecular phylogeny indicates that they represent closely related species (*Kroh & Mooi, 2024*) that split ∼6 million years ago (*Coppard & Lessios, 2017*). These two species are similar in size, mode of life, feeding, and habitat preferences. They are common in soft-bottom sandy habitats of the Gulf of Mexico and often occur sympatrically. Similarly to *C. subdepressus*, they are mobile, semi-infaunal to epifaunal detritivores. The fossil record of the two species dates back to the middle-to-late Pleistocene (*Coppard & Lessios, 2017*).

*Mellita tenuis* is a sand dollar (family Mellitidae) common in the eastern Gulf of Mexico, where it often occurs in dense congregations (*e.g.*, *Cleveland & Pomory, 2022*). The species is a mobile epifaunal to shallow infaunal detritivore that ingests sand-sized grains. The species is known from the Pleistocene of Florida (*Coppard, Zigler & Lessios, 2013*).

*Leodia sexiesperforata* is a sand dollar (family Mellitidae) that is widespread in the Gulf of Mexico. This species is a mobile, infaunal detritivore that occurs in shallow-water soft-bottom sandy habitats, often in dense congregations (*Grun & Kowalewski, 2022*). The species is known from the late Pleistocene fossil record of Florida (*Mooi & Peterson, 2000*; *Portell & Oyen, 2002*) and Jamaica (*Mitchell, James & Brown, 2006*).

## Environmental estimates (salinity and temperature)

The temperature estimates used here were downloaded from *NOAA (2024)* using seasonal datasets with the highest available resolution (0.1°). To derive the most relevant temperature estimates, the NOAA sectors nearest to our sampling sites were selected based on the nearest geographic distance calculated using R package 'geosphere' (*Hijmans, 2022*). The distances varied from ∼2 to ∼40 km. Because all sites represent shallow water settings with fully mixed waters, water temperatures were effectively depth invariant within sites and surface water temperatures can serve as proxies regardless of the site's depth. This assumption was confirmed by the analysis of the *NOAA (2024)* database: the temperature range in 0-to-25 m depth range averaged 0.05 °C and only 5% of NOAA sectors exceeded 1 °C difference within the top 25 m of the water column. Annual temperature was estimated as the arithmetic mean of the four seasonal means, while range of seasonal means was used as a measure of annual variability in temperature (Fig. 2; Data S2).

Salinity data were downloaded from *Gulf of Mexico Ocean Observation (GCOOS) (2019)*, an online-accessible aggregator of salinity data for the Gulf of Mexico. The GCOOS sites most proximal to our sampling sites were selected based on the nearest geographic distance calculated using R package 'geosphere' (*Hijmans, 2022*). Mean and median salinity values were calculated based on all measurements available for the most proximal GCOOS station (Data S2). The distance varied from ∼0.4 to ∼21 km. The variation in salinity was measured as the interquartile range of all available measurements.

## Bulk geochemical analyses

All analyzed specimens ($n = 43$) were live-collected individuals kept on ice and subsequently fixed in 95% scientific-grade isopropanol. Approximately 1/5 of each specimen (posterior sector cut through ambulacral lunules and running through the center of the petals, excluding lantern) (Fig. 3) were shortly bleached with sodium hypochlorite and split out into smaller fragments. These fragments were mechanically cleaned for the presence of soft tissue remnants, bleached again, and then washed with the aid of Milli-Q Water in the ultrasonic bath. These samples were finally dried at 50 °C, powdered and homogenized using an agate mortar and pestle. The resulting 43 aliquots (∼1 g) were analyzed separately for each specimen using the AQ250-EXT package (ultratrace aqua regia/ICP-AES and MS; Inductively Coupled Plasma Emission Spectrometry-Mass Spectrometry) at a commercial geochemical laboratory (Bureau Veritas Minerals). The entire set of analyzed elements
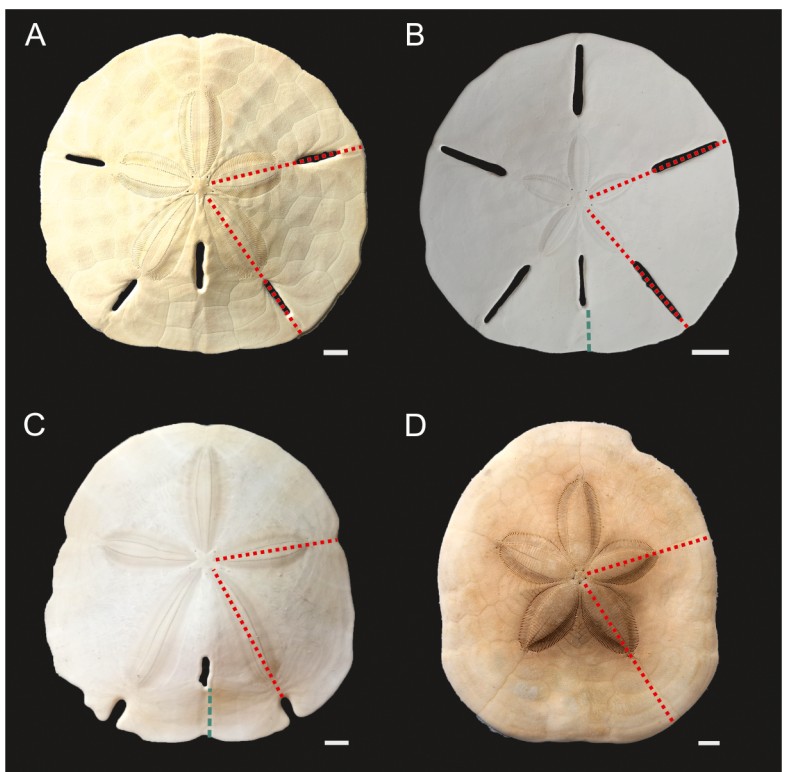

**Figure 3** **Echinoid taxa examined in this study.** (A) *Mellita tenuis*, (B) *Leodia sexiesperforata*. (C) *Encope* spp. (D) *Clypeaster subdepressus*. Red fine dotted lines indicate regions sampled for bulk ICP-AES and MS analyses. Green thick dotted lines show sections analysed *via* nanoindentation. Scale bars 1 cm.

($n = 53$), along with their detection limits, can be found in Data S3. However, in this paper, we focus on discussing a subset of elements (Mg, strontium (Sr), barium (Ba), sulfur (S), lithium (Li), lead (Pb), zinc (Zn), nickel (Ni), cobalt (Co), manganese (Mn), boron (B), cadmium (Cd), sodium (Na)) that have often been used as environmental proxies (*e.g.*, *Iglikowska et al., 2020*; *Ulrich et al., 2021*) and/or yielded estimates above the detection limits. All elements were converted to molar concentrations, and, as is commonly practiced, normalized to Ca concentration and expressed as element/Ca (mmol/mol). For the nanoindentation analysis described below, Mg/Ca ratios were reported using mol/mol ratios. Data quality was monitored by blind insertion of sample duplicates, internal reference materials, and the certified reference materials (including STD OREAS262 and STD BVGEO01). Analytical accuracy was within 2.5% of the value of the certified reference material for Mg, Sr, S, Na, P and within 5% of the standard value for Mn, Pb, Ba, Li, Cd, Cu and Zn.

Because the specimens were subject to destructive analyses, the remaining fragmentary material was not archived in a formal museum collection. However, it is available upon request from the first author.

## Nanoindentation and fine scale microprobe analyses

Two specimens of *Leodia sexiesperforata* and two specimens of *Encope michelini* were selected for nanoindentation analyses. One fragment of each test, representing the peripheral side posterior sector, were embedded in epoxy resin, cut perpendicularly to the test margin through the interambulacral plates, and ultra-polished. A nanoindenter (NanoTest Vantage; Micro Materials, Wrexham, UK) with a Berkovitch tip was used to determine nanomechanical properties (*i.e.,* nanohardness [H] = the material's resistance to permanent or plastic deformation at the nano-micro level, in GPa) of the stereom following *Oliver & Pharr (1992)*. The maximum load of 2 mN, resulting in the maximum penetration depth of ∼200 nm, was applied. Three microstructurally distinct regions from each specimen were analyzed separately, namely (1) the outermost solid, imperforate to perforate stereom layer (tubercles), (2) inner galleried stereom, and (3) inner massive and coarse labyrinthic stereom. For each stereom type, three microregions were analyzed resulting in nine separate analyses for each specimen (three analysis per region, three regions per specimen) for a total of 36 nanoscale analyses (Data S4, S5). Following the nanoindentation analysis, spot elemental measurements (Mg/Ca) in the same microregions were determined (Data S4, S5) with the aid of wavelength-dispersive spectroscopy (WDS) performed on a CAMECA SX100 electron microprobe at the Micro-Area Analysis Laboratory, Polish Geological Institute-National Research Institute in Warsaw, Poland. The following conditions were used: accelerating voltage: 15 kV; beam current: 5 nA for calcium and 20 nA for other elements; a beam diameter: ∼5 μm; standard: 'NIST' (serial number: 12570). Detection limit for Mg under this method was ∼0.01 wt %.

## Analytical methods

The relatively small sample sizes, which reflect prohibitive costs and the time-consuming nature of both nano-scale analyses and *in situ* collecting of often sparse echinoid populations *via* SCUBA, limits the statistical power and application of multi-factor models, which otherwise would have been highly appropriate here. Consequently, only simple descriptors and tests are feasible, including simple descriptors of central tendency, two-sample tests based on medians, bivariate correlation and semi-partial correlation analyses (partialling out interactions between salinity, temperature, and depth), and exploratory ordination methods. Correlation analyses (Pearson $r$) were limited to assessment of individual trace element ratios against extrinsic, independently derived environmental variables (salinity, temperature, depth). Semi-partial correlations (Pearson $r_{SP}$) were estimated separately for each trace element ratio against the three extrinsic environmental variables partialled out against each other. Semi-partial correlations were performed using the *R* package "pcor" (*Kim, 2015*).

Multivariate analyses included exploratory principal component analysis (PCA) analyses based on *z*-standardized data (*i.e.,* correlation PCA) to remove the effect of huge differences in variance across variables. PCA analysis was supplemented by correlation plots, loading plots, and bootstrap analysis of eigenvalues. The criterion 0.7*L, where L is a sum of proportional eigenvalues divided by number of variables (*i.e.,* mean proportional eigenvalue), was used to decide which principal components should be considered in the
interpretation (*Jolliffe, 2002*). It should be noted here that PCA is a parametric method that is not optimal for compositional data with variables affected by constant-sum-constraint (an issue further amplified here by Ca normalization). However, for high-dimensional data (as is the case here: 53 variables total) those effects tend to be less severe (*Kucera & Malmgren, 1998*). We selected PCA analysis mainly for comparative purposes as this method was previously employed for analyzing trace element ratios in echinoderm skeletons (*Iglikowska et al., 2020*). However, to evaluate the robustness of the results, the analysis was supplemented with nonmetric multi-dimensional scaling (NMDS). The NMDS analysis was performed on $z$-standardized variables using Euclidean-distance dissimilarity matrix and $k = 2$ dimensions (the stress <0.2 was used as our criterion for acceptable stress). NMDS was performed using the function metaMDS in the package 'vegan' (*Oksanen et al., 2024*). As shown below, the NMDS and PCA ordinations produced highly consistent ordination patterns.

In the case of nanoscale analyses, two-sample comparisons based on single factors (echinoid skeletal zone and echinoid species) were applied. Whereas single-factor analyses ignore confounding effects of other factors (*e.g.*, taxon), the balanced sampling design mitigates this issue. For example, in the case of nano-scale analyses, both taxa include the same set of measurements across the same set of three microstructural zones, and outer and inner zones have the same number of measurements for each taxon. In addition, in the case of the nano-scale analysis, the two taxa came from different localities/environments and thus effect 'taxon' may reflect environmental parameters or vital effects (a conundrum explicitly considered in the interpretations below). These issues impose limitations on interpretations of effects and significance tests. Pearson correlation coefficient was used to assess the strength of association between Mg/Ca ratios and nanohardness.

To ensure conservative statistical interpretations, a Bonferroni correction was applied when appropriate with significance alpha [$\alpha$] set to Bonferroni-corrected value of $\alpha_B = 0.05/k$, where k is the total number of tests of a given type performed in the study. All figures (except Fig. 3) and analyses were performed in R version 4.3.1 (*R Core Team, 2023*). R script is provided in Data S1 and custom functions for PCA used in the script are provided in Data S6, S7. Raw bulk geochemical data (not used in the analyses) are provided in Data S3. SEM images of echinoids showing sampling points for nanoscale analyses are included in Data S5.

# RESULTS

## Trace element analysis

Correlations between the analyzed element ratios, salinity and temperature varied across elements (Table 2; Fig. 4). However, salinity and temperature were strongly correlated with each other across the sites ($r = 0.84$, $p < 0.0001$). In addition, water depth was weakly associated with lower temperature ($r = -0.31$, $p = 0.04$), but not salinity ($r = -0.05$, $p = 0.74$). After partialling out interactions between salinity, temperature, and water depth (Table 2), no notable or significant semi-partial correlations are observed for temperature (Table 2). However, after accounting for the temperature and water depth

**Table 2** Pearson correlations and semi-partial correlation coefficients (accounting for interactions between temperature, salinity, and depth) for thirteen elemental ratios versus temperature, salinity, and depth.

| X/Ca ratio | Temperature | | Salinity | | Water depth | |
|---|---|---|---|---|---|---|
| | $r$ | $p$ | $r$ | $p$ | $r$ | $p$ |
| *correlation (Pearson)* | | | | | | |
| Mg | 0.64 | <0.00001** | 0.70 | <0.00001** | −0.04 | 0.800 |
| Sr | 0.49 | 0.0008** | 0.59 | 0.00003** | 0.15 | 0.345 |
| Ba | 0.07 | 0.661 | 0.01 | 0.967 | −0.31 | 0.041* |
| S | 0.35 | 0.023* | 0.67 | <0.00001** | 0.41 | 0.007* |
| Li | 0.45 | 0.003* | 0.72 | <0.00001** | 0.33 | 0.029* |
| Pb | 0.46 | 0.002* | 0.44 | 0.003* | 0.01 | 0.963 |
| Zn | 0.48 | 0.001** | 0.47 | 0.001** | −0.28 | 0.074^ |
| Mn | −0.38 | 0.013* | −0.21 | 0.166 | 0.23 | 0.147 |
| Na | −0.23 | 0.139 | −0.25 | 0.106 | 0.19 | 0.226 |
| B | 0.39 | 0.010^ | 0.52 | 0.0003** | 0.32 | 0.039* |
| P | 0.15 | 0.343 | 0.16 | 0.296 | −0.01 | 0.970 |
| Cd | −0.27 | 0.080^ | −0.31 | 0.046* | 0.26 | 0.089^ |
| Cu | 0.34 | 0.026* | 0.28 | 0.072^ | −0.04 | 0.809 |
| *semi-partial correlation (Pearson)* | | | | | | |
| Mg | 0.29 | 0.067^ | 0.30 | 0.053^ | 0.05 | 0.765 |
| Sr | 0.21 | 0.183 | 0.29 | 0.068^ | 0.20 | 0.209 |
| Ba | 0.03 | 0.833 | −0.08 | 0.610 | −0.22 | 0.166 |
| S | 0.02 | 0.907 | 0.55 | 0.0002** | 0.42 | 0.006* |
| Li | 0.15 | 0.353 | 0.50 | 0.0008** | 0.38 | 0.013* |
| Pb | 0.25 | 0.117 | 0.06 | 0.702 | 0.12 | 0.464 |
| Zn | 0.16 | 0.331 | 0.17 | 0.277 | −0.20 | 0.208 |
| Mn | −0.27 | 0.082^ | 0.16 | 0.326 | 0.03 | 0.847 |
| Na | −0.01 | 0.931 | −0.14 | 0.375 | 0.17 | 0.297 |
| B | 0.23 | 0.143 | 0.18 | 0.267 | 0.36 | 0.020* |
| P | 0.03 | 0.871 | 0.10 | 0.553 | −0.01 | 0.962 |
| Cd | 0.03 | 0.877 | −0.22 | 0.161 | 0.26 | 0.100 |
| Cu | 0.20 | 0.204 | −0.02 | 0.925 | 0.07 | 0.673 |

**Notes.**

Symbols: $r$ - Pearson correlation or semi-partial correlation coefficient, $p$ - significance value for the null hypothesis $r = 0$.
* $p$-values significant without Bonferroni correction.
^ $p$-values marginally significant without Bonferroni correction ($0.1 < p < 0.05$).
** $p$-values significant with the stringent Bonferroni correction ($p < 0.05/39 < 0.0013$).

effects, S and Li still displayed positive and significant or marginally significant semi-partial correlations with salinity (Fig. 4; Table 2) with a notable effect size ($r_{SP} > 0.5$). However, these correlations decrease and cease to be highly significant (Table 3) when salinity for site CK-5 is reassigned (see 'Material and Methods'). The subsequent analyses focus primarily on salinity (Fig. 4).

Comparisons of taxa within regions suggested that sympatric species from Florida Keys, where salinity was highest, differed in geochemical signatures for some elements. This was particularly notable for Mg/Ca and Li/Ca ratios in Florida Keys, with all *L. sexiesperforata*

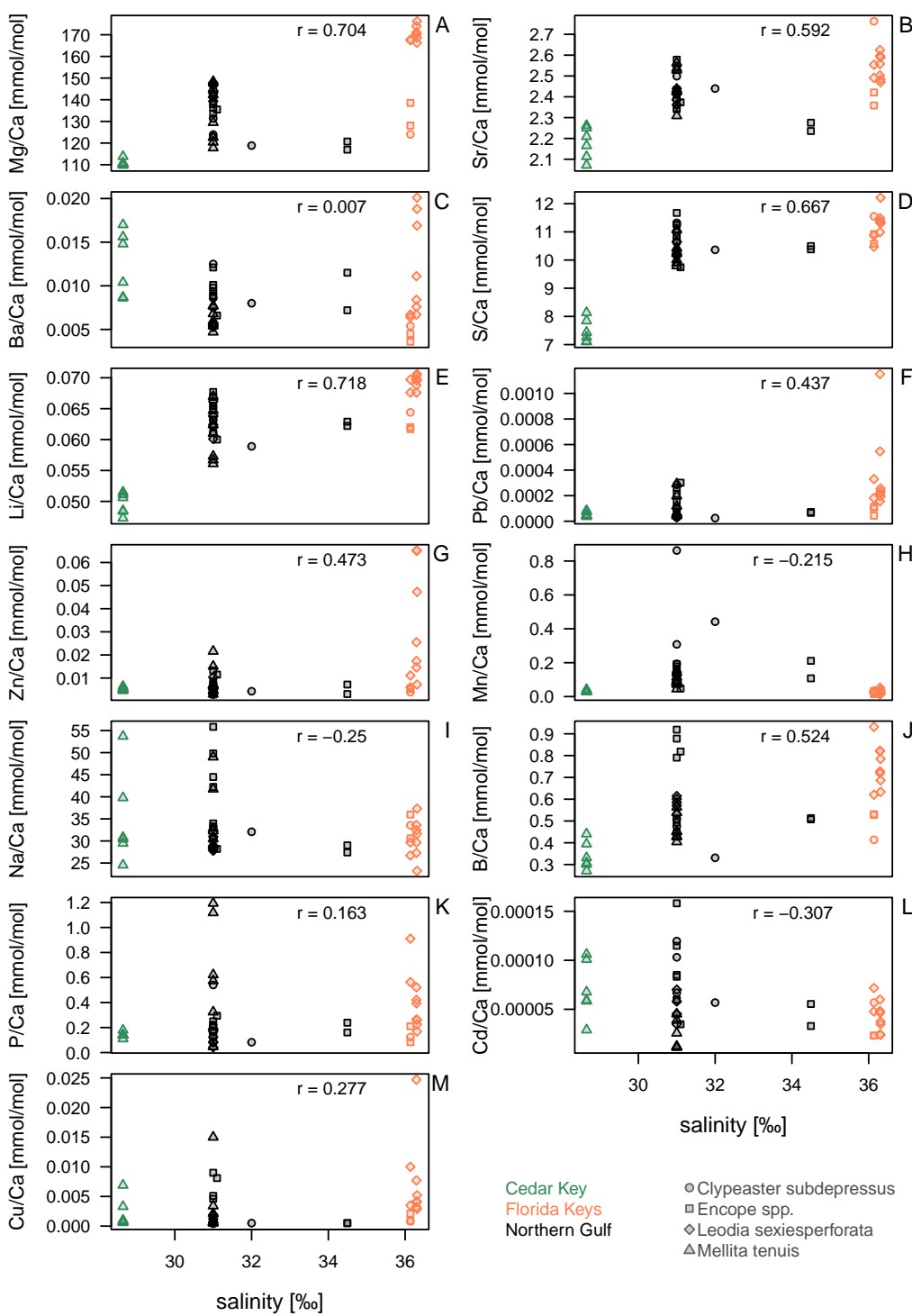

**Figure 4  Bivariate plots for element ratios obtained in the bulk geochemical analysis (Data S2) plotted against salinity.** Symbols represent individual specimens of echinoids. Symbols are color-coded by regions (see the left legend on the plot) and vary in shape depending on species to which a given specimen belongs (see the right legend on the plot). (A) Mg/Ca; (B) Sr/Ca; (C) Ba/Ca; (D) S/Ca; (E) Li/Ca; (F) Pb/Ca; (G) Zn/Ca; (H) Mn/Ca; (I) Na/Ca; (J) B/Ca; (K) P/Ca; (L) Cd/Ca; (M) Cu/Ca. Symbols: r - Pearson correlation coefficient.

**Table 3** Pearson correlations and semi-partial correlation coefficients (accounting for interactions between temperature, salinity, and depth) for thirteen elemental ratios *versus* temperature, salinity, and depth.

| X/Ca ratio | Temperature | | Salinity | | Water depth | |
|---|---|---|---|---|---|---|
| | *r* | *p* | *r* | *p* | *r* | *p* |
| *correlation (Pearson)* | | | | | | |
| Mg | 0.64 | <0.00001** | 0.64 | <0.00001** | −0.04 | 0.800 |
| Sr | 0.49 | 0.0008** | 0.51 | 0.0005** | 0.15 | 0.345 |
| Ba | 0.07 | 0.661 | 0.01 | 0.925 | −0.31 | 0.041* |
| S | 0.35 | 0.023* | 0.51 | 0.0005** | 0.41 | 0.007* |
| Li | 0.45 | 0.003* | 0.60 | 0.00002** | 0.33 | 0.029* |
| Pb | 0.46 | 0.002* | 0.43 | 0.004* | 0.01 | 0.963 |
| Zn | 0.48 | 0.001** | 0.47 | 0.001** | −0.28 | 0.074^ |
| Mn | −0.38 | 0.013* | −0.28 | 0.073^ | 0.23 | 0.147 |
| Na | −0.23 | 0.139 | −0.33 | 0.030* | 0.19 | 0.226 |
| B | 0.39 | 0.010* | 0.44 | 0.003* | 0.32 | 0.039* |
| P | 0.15 | 0.343 | 0.13 | 0.395 | −0.01 | 0.970 |
| Cd | −0.27 | 0.080^ | −0.33 | 0.031* | 0.26 | 0.089^ |
| Cu | 0.34 | 0.026* | 0.29 | 0.058^ | −0.04 | 0.809 |
| *semi-partial correlation (Pearson)* | | | | | | |
| Mg | 0.31 | 0.050 | 0.16 | 0.326 | 0.13 | 0.401 |
| Sr | 0.25 | 0.119 | 0.14 | 0.368 | 0.28 | 0.079^ |
| Ba | 0.07 | 0.682 | −0.12 | 0.437 | −0.23 | 0.148 |
| S | 0.15 | 0.349 | 0.28 | 0.072^ | 0.47 | 0.002* |
| Li | 0.21 | 0.193 | 0.30 | 0.059^ | 0.45 | 0.003* |
| Pb | 0.24 | 0.132 | 0.04 | 0.823 | 0.14 | 0.381 |
| Zn | 0.16 | 0.327 | 0.13 | 0.424 | −0.18 | 0.264 |
| Mn | −0.23 | 0.143 | 0.10 | 0.517 | 0.07 | 0.669 |
| Na | 0.08 | 0.607 | −0.26 | 0.096 | 0.22 | 0.173 |
| B | 0.28 | 0.081^ | 0.03 | 0.842 | 0.43 | 0.005* |
| P | 0.05 | 0.768 | 0.05 | 0.768 | 0.02 | 0.904 |
| Cd | 0.07 | 0.682 | −0.26 | 0.104 | 0.27 | 0.084^ |
| Cu | 0.18 | 0.247 | −0.01 | 0.950 | 0.07 | 0.667 |

**Notes.**
The same analysis as in Table 2, but with the salinity estimate for site CK-5 based on estimates for site CK-6. See text and the footnote in Table 1. Symbols: *r* - Pearson correlation or semi-partial correlation coefficient, *p* - significance value for the null hypothesis $r = 0$.

* *p*-values significant without Bonferroni correction.
^ *p*-values marginally significant without Bonferroni correction ($0.1 < p < 0.05$).
** *p*-values significant with the stringent Bonferroni correction ($p < 0.05/39 < 0.0013$).

specimens having much higher values when compared to *Encope* spp. and *C. subdepressus* specimens (Figs. 4A, 4E). Similar, if less pronounced offsets were observed for Ba/Ca (Fig. 4C), Pb/Ca (Fig. 4F), Zn/Ca (Fig. 4G), B/Ca (Fig. 4J), and Cu/Ca (Fig. 4M). In contrast, the offsets in those geochemical signatures were notably smaller or absent in the case of specimens collected from the lower salinity setting of the Northern Gulf.

**Table 4 Intraspecific comparisons of trace element ratios for *Leodia sexiesperforata* from two regions with different salinity regimes.**

| Trace element ratio | Florida keys | Northern Gulf | p | Offset | Standardized offset |
|---|---|---|---|---|---|
| Mg/Ca (mmol/mol) | 170.06 | 146.49 | 0.001** | **23.57** | **0.15** |
| Sr/Ca (mmol/mol) | 2.55 | 2.39 | 0.001** | **0.17** | **0.07** |
| Ba/Ca (mmol/mol) | 0.0084 | 0.0055 | 0.003** | **0.0029** | **0.42** |
| S/Ca (mmol/mol) | 11.34 | 10.64 | 0.012* | **0.70** | **0.06** |
| Li/Ca (mmol/mol) | 0.0696 | 0.0638 | 0.003** | **0.0058** | **0.09** |
| Pb/Ca (mmol/mol) | 0.000233 | 0.000038 | 0.001** | **0.0002** | **1.44** |
| Zn/Ca (mmol/mol) | 0.0174 | 0.005 | 0.007* | **0.0124** | **1.11** |
| Mn/Ca (mmol/mol) | 0.0237 | 0.129 | 0.001** | −0.11 | −1.38 |
| Na/Ca (mmol/mol) | 29.68 | 28.28 | 0.518 | 1.39 | 0.05 |
| B/Ca (mmol/mol) | 0.73 | 0.59 | 0.001** | **0.14** | **0.22** |
| P/Ca (mmol/mol) | 0.39 | 0.08 | 0.002** | **0.31** | **1.3** |
| Cd/Ca (mmol/mol) | 0.000046 | 0.000045 | 0.797 | 0.000001 | 0.02 |
| Cu/Ca (mmol/mol) | 0.0041 | 0.0012 | 0.003** | **0.0029** | **1.09** |

**Notes.**

Median elemental ratio values are reported separately for each region. Statistical significance ($p$) estimated using non-parametric two-sample Wilcoxon rank test. The offset estimated by the difference between medians of the regions, with positive values indicating higher median ratio for Florida Keys specimens and negative values indicating higher median ratio for Northern Gulf specimens. Standardized offset estimated as offset divided by midpoint of the two median values. The values set in bold print indicate trace element ratios that are significantly higher (with or without Bonferroni correction) for Florida Keys (the higher salinity region).

*$p$-values significant without Bonferroni correction.

**$p$-values significant with the stringent Bonferroni correction ($p < 0.05/13 < 0.0038$).

Within-species comparisons across different regions suggest consistent offsets in multiple element ratios. *L. sexiesperforata* from Florida Keys displayed higher trace element ratios (except for Mn/Ca ratio) when compared to conspecific specimens from the lower salinity settings in Northern Gulf, and in many cases those offsets were significant (Table 4). Similarly, *M. tenuis* displayed consistently higher Mg/Ca (Fig. 4A), Sr/Ca (Fig. 4B), S/Ca (Fig. 4B), Li/Ca (Fig. 4E), and Pb/Ca (Fig. 4K) ratios in Northern Gulf compared to the lower salinity sites at Cedar Key (although a formal analysis could not be carried out due to small sample size and uncertainty related to the salinity estimates for the site CK-5). In some cases, high intraspecific variation was observed within the same region/salinity regime, especially for Ba/Ca and Zn/Ca ratios in *L. sexiesperforata* (Figs. 4C, 4G).

Despite confounding effects of interspecific and intraspecific variability, when data are pooled across taxa and the specimens from the high-salinity region of Florida Keys is compared to specimens from the lower salinity regimes of Northern Gulf and Cedar Key (combined), the majority of trace element ratios are higher for Florida Key specimens (Table 5), including Mg/Ca (Fig. 4A), Sr/Ca (Fig. 4B), S/Ca (Fig. 4D), Li/Ca (Fig. 4E), Pb/Ca (Fig. 4F), Zn/Ca (Fig. 4G), B/Ca (Fig. 4J), and Cu/Ca (Fig. 4M). The Mn/Ca ratios are a notable exception showing a reverse trend (Fig. 4H; Table 5).

Principal component analyses of $z$-standardized elemental ratios produced an ordination consistent with correlation analyses by highlighting the distinctness of environmental settings (the three regions) and the presence of inter- and intraspecific variation within

**Table 5  Inter-regional comparison of trace element ratios.** Data pooled across all taxa. The Northern Gulf and Cedar Key specimens were pooled together to represent the lower salinity "Gulf" data to be contrasted against the high salinity Florida Key specimens. Median elemental ratio values are reported separately for each region. Statistical significance ($p$) estimated using non-parametric two-sample Wilcoxon rank test. The offset estimated by the difference between medians of the regions, with positive values indicating higher median ratio for Florida Keys specimens and negative values indicating higher median ratio for Gulf specimens. Standardized offset estimated as offset divided by midpoint of the two median values. The values set in bold print indicate trace element ratios that are significantly higher (with or without Bonferroni correction) for Florida Keys (the higher salinity region).

| Trace Element Ratio | Florida keys (median) | Gulf (median) | $p$ | Offset | Standardized offset |
|---|---|---|---|---|---|
| Mg/Ca (mmol/mol) | 168.21 | 131.37 | 0.00004[**] | **36.84** | **0.25** |
| Sr/Ca (mmol/mol) | 2.53 | 2.39 | 0.0004[**] | **0.14** | **0.06** |
| Ba/Ca (mmol/mol) | 0.01 | 0.01 | 0.85 | −0.001 | −0.11 |
| S/Ca (mmol/mol) | 11.32 | 10.33 | 0.00007[**] | **1.00** | **0.09** |
| Li/Ca (mmol/mol) | 0.069 | 0.062 | 0.0001[**] | **0.0070** | **0.11** |
| Pb/Ca (mmol/mol) | 0.00020 | 0.00006 | 0.002[**] | **0.00014** | **1.06** |
| Zn/Ca (mmol/mol) | 0.013 | 0.006 | 0.007[*] | **0.007** | **0.79** |
| Mn/Ca (mmol/mol) | 0.024 | 0.116 | 0.000002[**] | −0.092 | −1.32 |
| Na/Ca (mmol/mol) | 31.09 | 30.81 | 0.58 | 0.28 | 0.01 |
| B/Ca (mmol/mol) | 0.70 | 0.49 | 0.002[**] | **0.21** | **0.35** |
| P/Ca (mmol/mol) | 0.26 | 0.17 | 0.076[^] | 0.093 | 0.43 |
| Cd/Ca (mmol/mol) | 0.00004 | 0.00006 | 0.14 | −0.00002 | −0.32 |
| Cu/Ca (mmol/mol) | 0.00380 | 0.00080 | 0.005[*] | **0.0030** | **1.30** |

**Notes.**
[*] $p$-values significant without Bonferroni correction.
[^] $p$-values marginally significant without Bonferroni correction ($0.1 < p < 0.05$).
[**] $p$-values significant with the stringent Bonferroni correction ($p < 0.05/13 < 0.0038$).

environmental settings (Fig. 5A). The first two principal components accounted for 51% of variance in the data and variables were moderately or strongly correlated either with PC1 or PC2, or both components (Fig. 5B). The long arrows that approach the outer blue circle ('circle of correlation'; *sensu Abdi & Williams, 2010*) indicate the variables for which almost all variance was captured by the first two principal components. Conversely, the loading plot indicated that the variance explained by the first two components was partitioned relatively evenly across all variables (Fig. 5C). The bootstrap analysis of eigenvalues suggested that the first four principal components may be potentially informative (Fig. 5D). The correlation plot (Fig. 5B) indicated that specimens with high PC1 scores tended to have higher elemental ratios for all elements except Ba and Mn that were uncorrelated with PC1 scores. Specimens with high PC2 scores had low Ba/Ca and high Mn/Ca scores. PC2 scores were also negatively correlated with Pb/Ca and Zn/Ca ratios and showed less pronounced positive relationships with S/Ca, Sr/Ca, and Li/Ca ratios (Fig. 5B). Among the external variables, salinity displayed strong correlation with PC1 ($r = -0.74$, $p < 0.0001$). Temperature also displayed notable correlation with PC1 ($r = 0.61$, $p < 0.0001$), but based on semi-partial correlation analysis (Tables 2 and 3), this correlation reflected co-variation

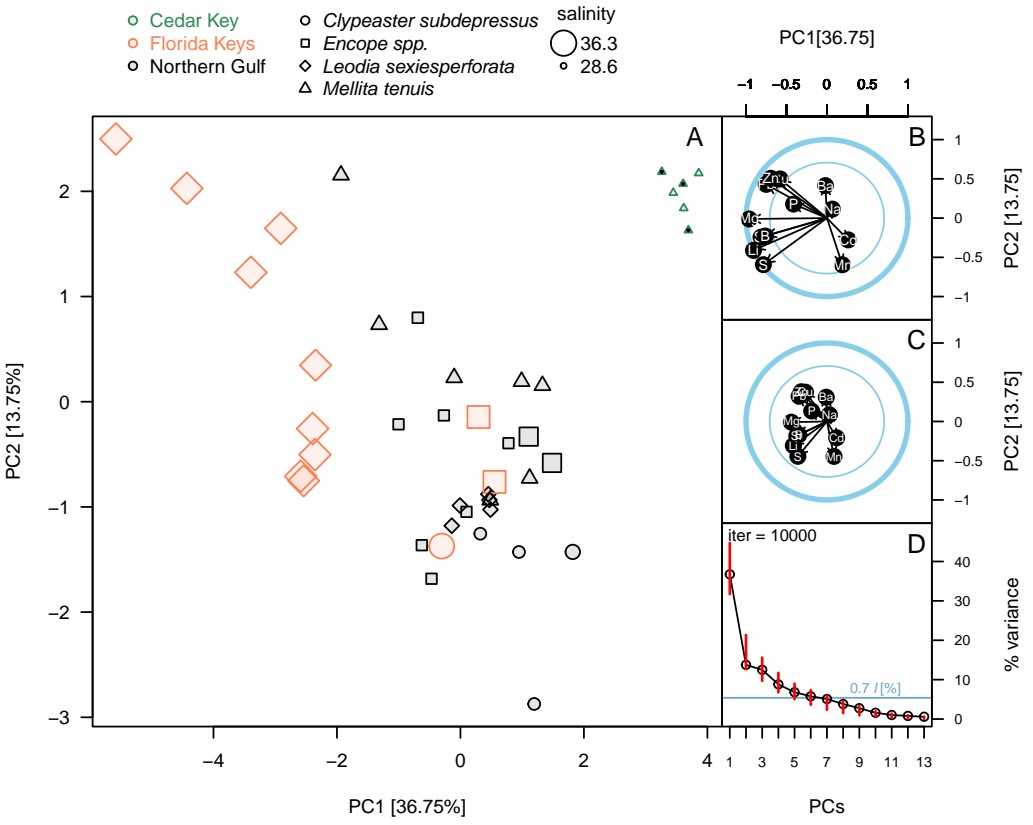

**Figure 5 Principal component analysis of the elemental ratios obtained in the bulk geochemical analysis.** All values were z-standardized prior to analysis (correlation PCA). (A) PCA ordinations of the first two principal components. Symbols, color-coded by region, representing individual specimens. Specimens of *Mellita tenuis* from the site CK-5, which could be alternatively classified as a "Northern Gulf" site with higher salinity, are marked by additional black dots inside the symbols (green triangles). Percent variance accounted for by each PC1 indicated in axes labels. All symbols are scaled by salinity estimates for a site from which a given specimen was collected. Larger symbols indicate specimens collected from a relatively higher salinity regime. (B) Correlation plot indicating correlations of individual elemental ratio variables with the first two principal components. Longer arrows indicate variables with the higher amount of variance explained by the first two PC axes. Thicker circle indicates a circle of correlation (arrows terminating on the circle indicate variables for which the variance is fully accounted for by the first two components). The inner circle denotes 50% of variance. (C) Loading plot indicating how variance of the first two PCs is partitioned across all elemental ratio variables. (D) Bootstrap estimates of 95% confidence intervals around eigenvalues (red bars) expressed as the total proportion of variance. The dark blue horizontal line indicates *0.7*l* criterion for considering the informative value of a given principal component. The value *0.7*l* value is expressed as the percentage of the total variance. Because *l* = 1 and total variance = 13 (*i.e.,* number of variables), the percent value is 100*(0.7/13) = 5.38%. Principal components with red bars above the blue line are considered to have variance significantly higher than is predicted by the *0.7*l*.

of temperature with salinity. All other correlation coefficients of environmental variables with PC1 and PC2 were <0.5 thus accounting at most for 25% of variance in the PC scores.

The ordination (Fig. 5A) indicated that the three regions were distinct in geochemical signature, with the three environmental settings largely separating along PC1 indicating that most element ratios increased with salinity. High intra-regional, intra-specific

variability was observed for *L. sexiesperforata* from Florida Keys. Intra-regional interspecific differences in element ratios were evident in multiple cases, including differences between *L. sexiesperforata* and *Encope* spp. in Florida Keys and *M. tenuis* and *C. subdepressus* in Cedar Key (Fig. 5A). Specimens of the same species from different regions differed in geochemical signatures in the case of *M. tenuis* and *L. sexiespeforata* (Fig. 5A). In contrast, all five specimens of *C. subdepressus* plotted in the upper central part of the ordination indicating relatively comparable geochemical signatures regardless of the environmental setting.

Whereas PC3 and PC4 axes may be potentially informative (Fig. 5D), ordination plots of those higher components (Figs. 6B and 6C) do not indicate any notable changes to patterns observed in the first two dimensions (Figs. 5A or 6A). The NMDS ordination (Fig. 6D) is remarkably consistent with the PCA ordination indicating that the results are reproducible using multivariate ordination methods that are methodologically disparate.

A comparison of Mg/Ca and S/Ca, illustrates all above mentioned patterns for the two element ratios (Fig. 7), highlighting consistent interregional offsets within species (*M. tenuis* for Northern Gulf *versus* Cedar Key and *L. sexiesperforata* for Florida Keys *versus* Northern Gulf), high within-species within-region variability (*L. sexiesperforata* in Florida Keys), interspecific differences within regions (*L. sexiesperforata versus* other species in Florida Keys), and intraspecific invariance across regions with different salinities (*Encope* spp. and *C. subdepressus* showing similar values for Florida Keys and Northern Gulf).

Although there is a substantial variation in body size, both among and within the four echinoid taxa, the body size was not correlated notably or significantly with element ratios after accounting for interactions with environmental variables (semi-partial correlation $r \le 0.3$, $p > 0.05$, in all cases). Consistently with those results, specimens in the PCA ordinations (Fig. 5) did not ordinate according to size. For example, the two species with smaller body sizes (*M. tenuis* and *L. sexiesperforata*) plotted on the opposite ends of PC1 axis and were distributed widely along PC2 axis (Fig. 5). Body size was not correlated notably or significantly with PC1 ($r = 0.28$, $p = 0.10$) and showed low correlation with PC2 scores ($r = -0.36$, $p = 0.03$).

## Micro- and nano-scale analysis

A total of 36 nano-scale analyses (Data S4), with simultaneous nano-scale measurements of Mg/Ca mmol/mol ratios (Mg) and hardness (H) were obtained for four echinoid tests (nine analyses per specimen), including two specimens of *Encope michelini* (one from Northern Gulf and one from Florida Keys) and two specimens of *Leodia sexiesperforata* (both from Florida Keys), which were also used for bulk geochemical analyses.

When data were pooled across the two analyzed taxa, notable and statistically significant differences were observed in all comparisons between the nano-scale measurements for the outer and inner zones of echinoid tests (Fig. 8). The imperforate stereom of the outer zone was characterized by significantly elevated Mg/Ca and H values when compared to the thin galleried and coarse labyrinthic stereom of the inner zone: (1) Mg/Ca: median (outer) = 0.193, median (inner) = 0.168, $W = 59.5$, $p = 0.005$; (2) H: median (outer) = 4.54, median (inner) = 3.98, $W = 15.5$, $p = 0.00002$ ($\alpha_B = 0.05/6 = 0.008$; two-sample
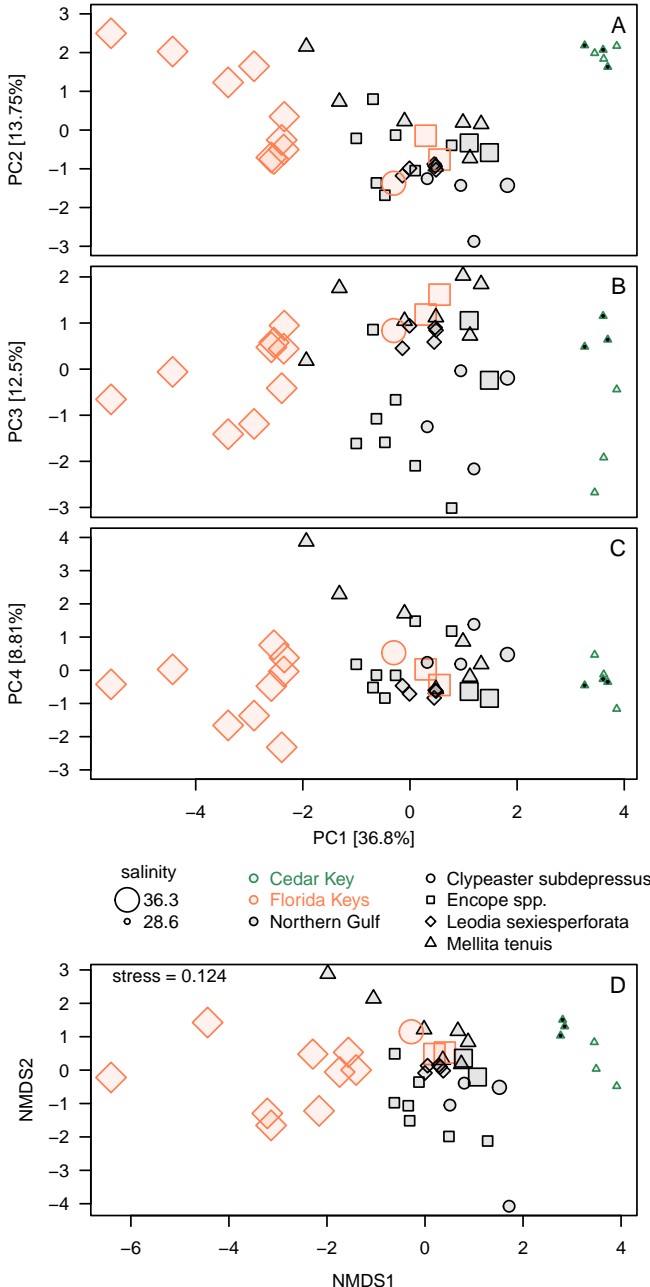

**Figure 6 Additional ordination plots.** (A–C) PCA ordination plots for the first four principal components based on the elemental ratios obtained in bulk geochemical analysis with values $z$-standardized prior to analysis (correlation PCA). (A) Ordination plot for PC1 and PC2 (the same as Fig. 5A). (B) Ordination plot for PC1 and PC2 (the same as Fig. 5A). (C) Ordination plot for PC1 and PC2 (the same as Fig. 5A). (D) Nonmetric Multi-Dimensional Scaling (NMDS) for $k = 2$ dimensions based on dissimilarity matrix of Euclidean distances for $z$-standardized trace element ratios. In all PCA and NMDS plots, symbols represent individual specimens color-coded by region. Specimens of *Mellita tenuis* from the site CK-5, which could be alternatively classified as a "Northern Gulf" site with higher salinity, are marked by additional black dots inside the symbols (green triangles). All symbols are scaled by salinity estimates for a site from which a given specimen was collected. Larger symbols indicate specimens collected from a relatively higher salinity regime.

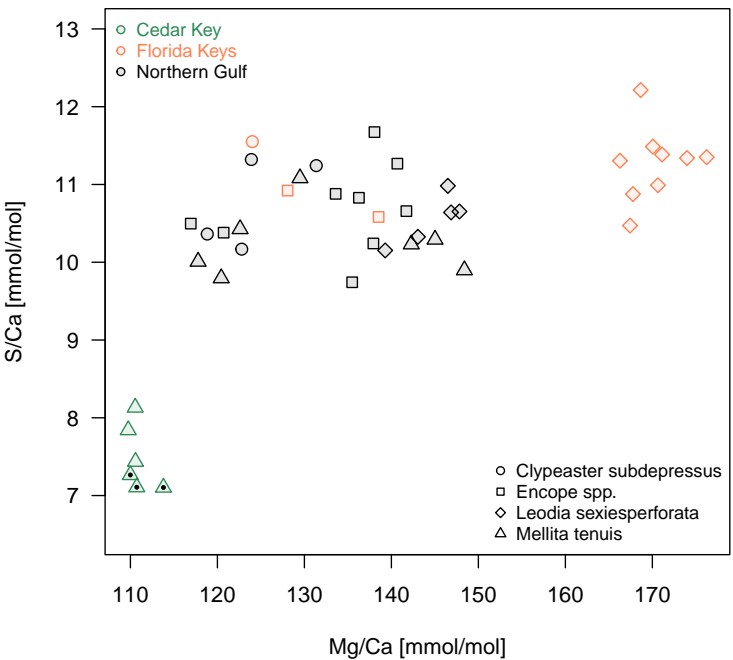

**Figure 7** **A bivariate plot of Mg/Ca and S/Ca ratios.** Symbols, with symbol shapes indicating taxa and colors denoting regions, represent individual specimens. Specimens of *Mellita tenuis* from the site CK-5, which could be alternatively classified as a "Northern Gulf" site with higher salinity, are marked by additional black dots inside the symbols (green triangles).

Wilcoxon rank test). The median values of Mg/Ca and H were also higher for the outer zone compared to the inner zone when data were split by taxon (Table 6).

The comparison of two taxa (Table 6) for all data indicated that median Mg was significantly higher for *L. sexiesperforata* than for *E. michelini* (median (*L. sexiesperforata*) = 0.192, median (*E michelini*) = 0.158, $W = 25$, $p = 0.0002$, $\alpha_B = 0.05/6 = 0.008$; two-sample Wilcoxon rank test). In contrast, median values of H were indistinguishable statistically between the two taxa: (1) H: median (*L. sexiesperforata*) = 4.25, median (*E. michelini*) = 4.03, $W = 149$, $p = 0.69$. The consistent differences in Mg between the two taxa persisted when per-taxon median values were computed for each zone separately, with higher median values observed for *L. sexiesperforata* when compared to *E. michelini* for both the inner and outer zones (Table 6).

Mg/Ca ratios and nanohardness H were positively correlated, with moderate-to-high correlation coefficient (Fig. 9; Table 7). For pooled data and for each taxon separately, these correlations were highly significant at Bonferroni-corrected significance level ($\alpha_B = 0.05/9$ = 0.0055; Table 7). In the case of *E. michelini*, the values obtained for specimens from Northern Gulf appeared systematically offset toward lower Mg/Ca ratios, but nanohardness was comparable for the two specimens.

Univariate analyses (Table 6, Fig. 8) and correlation analyses (Table 7, Fig. 9) summarized above consistently indicated that Mg and H were elevated for the outer zone of the echinoid test when compared to the inner zone. Concurrently, there was a consistent offset in Mg/Ca

**Table 6** A univariate summary for Mg/Ca and H nano-scale measurements grouped by skeletal zone (inner and outer test), taxon (*Encope michelini* and *Leodia sexiesperforata*), and taxon-zone combination.

| | Median | Mean | Standard deviation | Number of observations |
|---|---|---|---|---|
| *Mg/Ca* (mol/mol) | | | | |
| Inner (all data) | 0.168 | 0.169 | 0.023 | 24 |
| Outer (all data) | 0.193 | 0.195 | 0.019 | 12 |
| E. michelini (all data) | 0.158 | 0.160 | 0.019 | 18 |
| L. sexiesperforata (all data) | 0.192 | 0.195 | 0.015 | 18 |
| E. michelini (inner) | 0.152 | 0.149 | 0.009 | 12 |
| L. sexiesperforata (inner) | 0.190 | 0.189 | 0.010 | 12 |
| E. michelini (outer) | 0.185 | 0.183 | 0.010 | 6 |
| L. sexiesperforata (outer) | 0.200 | 0.207 | 0.017 | 6 |
| *H* | | | | |
| Inner (all data) | 3.980 | 3.992 | 0.241 | 24 |
| Outer (all data) | 4.540 | 4.515 | 0.189 | 12 |
| E. michelini (all data) | 4.030 | 4.139 | 0.385 | 18 |
| L. sexiesperforata (all data) | 4.250 | 4.193 | 0.285 | 18 |
| E. michelini (inner) | 3.955 | 3.920 | 0.228 | 12 |
| L. sexiesperforata (inner) | 4.020 | 4.063 | 0.241 | 12 |
| E. michelini (outer) | 4.575 | 4.578 | 0.206 | 6 |
| L. sexiesperforata (outer) | 4.510 | 4.452 | 0.164 | 6 |

ratio between the two taxa, with Mg values elevated for *L. sexiesperforata* when compared to *E. michelini* (Figs. 8–9). However, a joint comparison of the offsets between inner and outer zones for *L. sexiesperforta*, and *E. michelini* suggested that while the two taxa differed in Mg/Ca ratios, the nano-physical properties H were very similar (Fig. 10). The nanohardness H values of the inner and outer zones were nearly identical for the two taxa despite notable interspecific differences in Mg/Ca ratios, and the magnitude of the offset in H was also comparable between the two taxa (Fig. 8). Bootstrap-based confidence intervals around medians indicated substantial uncertainty (Figs. 8–10), but this uncertainty is not sufficient in magnitude to nullify those differences.

## DISCUSSION

Because of the logistically-imposed data limitations, the results reported here should be viewed as tentative and primarily serve as a guideline toward future studies by highlighting multiple intriguing outcomes with variable statistical support.

For element ratios of Ba/Ca, Pb/Ca, Cd/Ca, Cu/Ca, Zn/Ca, P/Ca and Mn/Ca, bulk geochemical analyses pooled across all taxa and environments revealed a heterogeneous distribution without any discernible patterns (Fig. 4). Previous studies on some echinoderm species have demonstrated that the bioaccumulation of some of these elements can be directly related to their concentrations in the environment (seawater, sediment and/or food), and thus they may serve as useful bioindicators of temporal variations

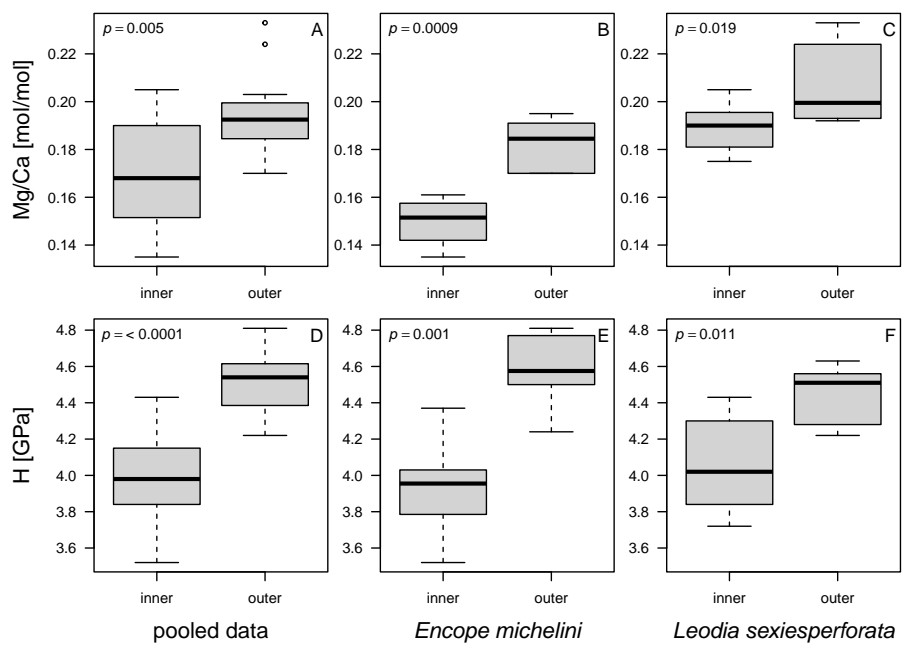

**Figure 8** **Box and whisker plots comparing Mg and H estimates in nano-scale analyses between inner and outer zones of echinoid tests.** (A–C). Mg/Ca ratios. (D–F). H (nanohardness) estimates. (A and D) Pooled data. (B and E) *Encope michellini*. (C and F) *Leodia sexiesperforata*. Symbols: *p* - significance value for a two-sample non-parametric Wilcoxon test.

in environmental contaminants (*e.g.*, *Temara et al., 1997*; *Temara et al., 1998*; *Gorzelak et al., 2017*; *Iglikowska et al., 2020*). Likewise, Na/Ca and B/Ca ratios showed no obvious trends, which may be somewhat surprising because these element ratios in biogenic calcite (mostly in foraminifers) were previously proposed as good proxies for salinity (*e.g.*, *Allen et al., 2011*; *Wit et al., 2013*). However, other studies indicated that these ratios are not straightforward salinity proxies because they can also be sensitive to other environmental variables (*e.g.*, *Henehan et al., 2015*; *Gray et al., 2023*). Nevertheless, the intraspecific and inter-regional comparisons of trace element ratios reported in this study from two regions with different salinity regimes revealed statistically higher trace element ratios (including B/Ca) than those from the lower salinity settings (Tables 4 and 5).

For Mg, Sr, S and Li, a systematic coupling of these elements at least for some species is notable, which suggest a simultaneous uptake and incorporation of these elements during skeleton formation. The results suggest that both physiological and environmental factors may be influencing geochemical signatures of echinoid tests. Extrinsic controls are suggested by differences within some species between regions (*i.e.,* increased ratios of these elements in *M. tenuis* and *L. sexiesperforata* from regions with increased salinity), similarities between species within some regions (especially the Northern Gulf), and overall differences in bulk geochemical compositions observed in multivariate ordinations (Figs. 5 and 6). In contrast, the lack of differences between regions within some species (*i.e.*, *Encope* spp. and *C. subdepressus* both display high variability in geochemical signatures) and distinct

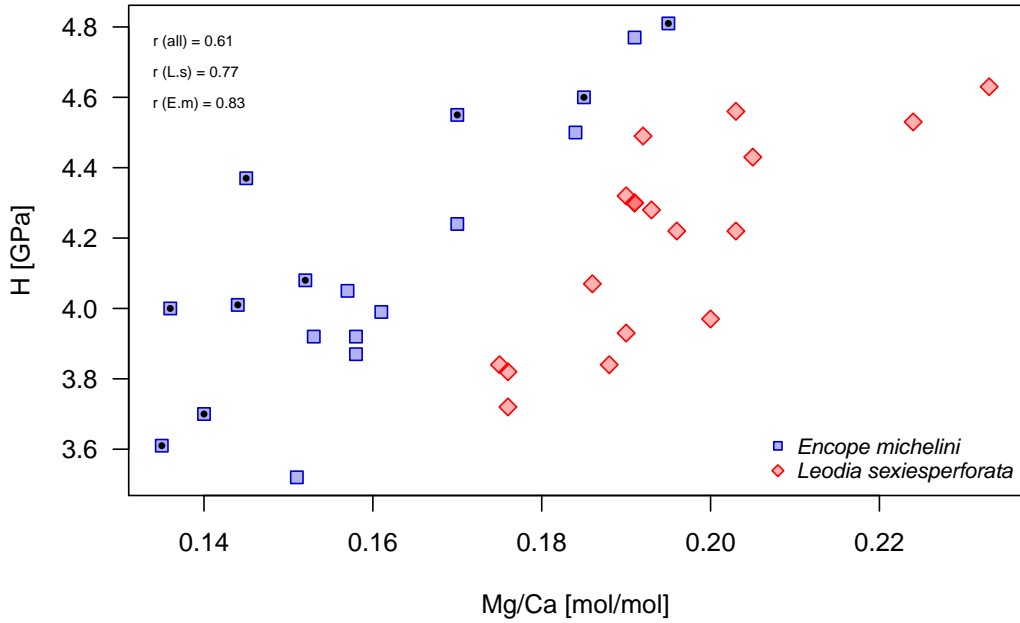

**Figure 9** **Bivariate plot of Mg/Ca ratios and H (nanohardness).** Symbols, color-coded by species, represent individual nano-scale measurements. *Encope michelini* measurements for one specimen from Northern Gulf are marked by black dots inside the symbols. All other measurements are for specimens from the Florida Keys. Symbols: $r$ (all) - Pearson correlation coefficient for pooled data; $r$ (E.m) - *Encope michelini* only; $r$ (L.s) - *Leodia sexiesperforata* only.

geochemical signatures of sympatric species (*L.* sexiesperforata *vs. Encope* spp. in Florida Keys) are indicative of species-intrinsic physiological effects. Considered jointly, these findings are consistent with some previous studies (*e.g.*, *Borremans et al., 2009*; *Hermans et al., 2010*; *Smith et al., 2016*; *Iglikowska et al., 2020*) documenting that both environmental and intrinsic biological (vital) factors can influence geochemical signatures to various degrees depending on the species and environment. In particular, at high-salinity regimes, some of the analyzed species exerted a differential vital effect on element fractionation during the stereom formation, but this physiological control over element such as Mg and Sr appears to be stronger for some taxa (*Encope* spp. and *C. subdepressus*), but less pronounced (or perhaps marginal) for other (*M. tenuis* and *L. sexiesperforata*).

The differences in elemental ratios among the four species are consistent with experimental studies that revealed different patterns in element fractionation in various echinoderm species. For instance, *Borremans et al. (2009)* observed a positive linear relationship between Sr/Ca in asteroid skeletons and salinity (0.94–1.69 (mmol/mol)/psu, *i.e.*, ~1% to ~1.5%/psu), the estimates similar in magnitude to the salinity effect observed for *M. tenuis* and *L. sexiesperforata*; Fig. 4). Similarly, *Hermans et al. (2010)*, demonstrated that the skeletal strontium/calcium (Sr/Ca) ratios in an echinoid species *Paracentrotus lividus* were linked to salinity. In contrast, *Pilkey & Hower (1960)* noted that the skeletal Sr/Ca ratios in tests of the echinoid genus *Dendraster* appeared unaffected by salinity. Similarly, the temperature effects on the skeletal Mg/Ca ratios may vary across echinoderm

**Table 7** Pairwise correlation coefficients and associated significance tests for the two nano-scale variables (Mg and H) obtained concurrently for 36 nano-sites.

| | r | 95% confidence intervals | | n | p |
| --- | --- | --- | --- | --- | --- |
| | | Lower limit | Upper limit | | |
| Pooled data - Mg *vs* H | 0.611 | 0.353 | 0.782 | 34 | 0.00008[*] |
| *E. michelini* - Mg *vs* H | 0.830 | 0.593 | 0.935 | 16 | 0.00002[*] |
| *L. sexiesperforata* - Mg *vs* H | 0.769 | 0.472 | 0.909 | 16 | 0.00019[*] |

**Notes.**

Symbols: *r* - Pearson correlation coefficient, *n* - sample size, and *p* - significance value.

[*]Statistically significant correlations at Bonferroni-corrected significance level, $\alpha_B = 0.05/3 = 0.0166$.

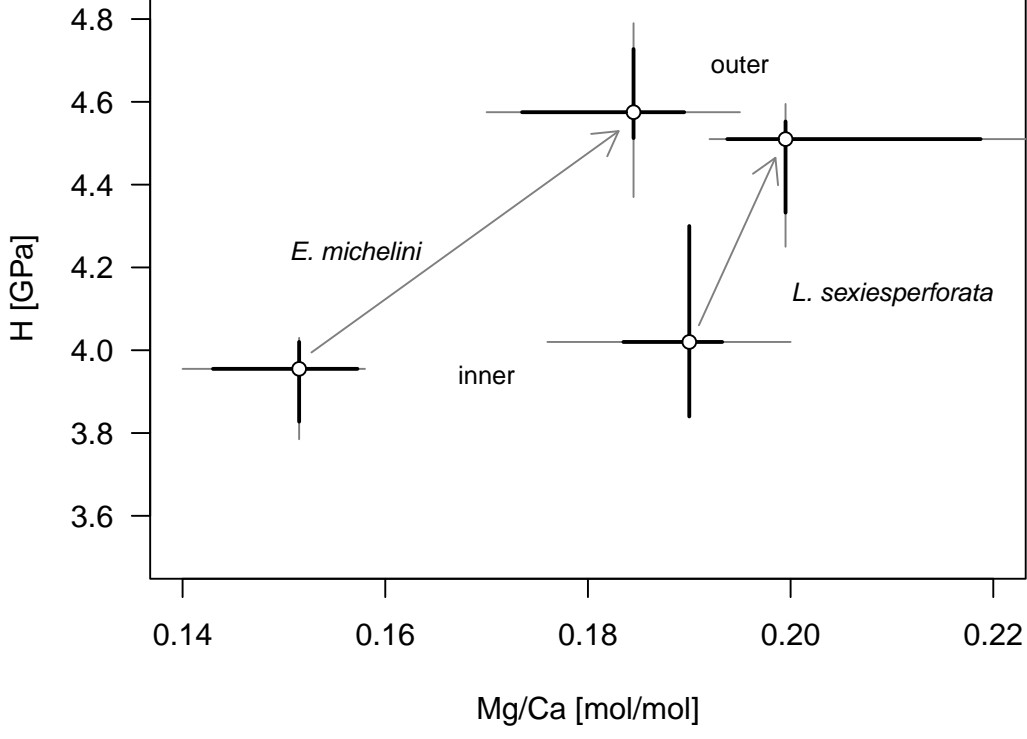

**Figure 10** Comparative bivariate plot of Mg/Ca ratios and H (nano-hardness) of *Encope michelini* and *Leodia sexiesperforata*. Gray dots indicate median values for inner and outer zones with Bootstrap 95% confidence intervals indicated by thin black lines. Bootstrap confidence intervals estimated using the function MedianCI, package DescTools (*Signorell, 2024*). Arrows indicate offsets between the inner and outer zones shown separately for the two taxa.

species (*e.g.*, *Hermans et al., 2010*; *Duquette et al., 2018*). The variable effects of salinity on different echinoid taxa from the same system are thus consistent with previous studies indicating that different species respond differently to salinity changes. Overall, these species-specific fractionation patterns impose limitations on the use of echinoderms (as a whole) in paleoenvironmental reconstructions, suggesting that salinity estimates based on skeletal element ratios may not be possible for all echinoderm species.

Our results showed that not only Mg and Sr but also, in particular, S and Li can be positively linked to salinity in some species. In the case of sulfur, this can be a spurious relationship because the concentration of sulfur, partly associated with the sulphated fraction of the organic matrix, tends to correlate with magnesium (*Gorzelak et al., 2013*). However, previous studies also suggested that biological control on the sulfur uptake in echinoderms may potentially override environmental influence (*Iglikowska et al., 2020*). In the case of Li, some biological control during Li incorporation into the skeletons of some invertebrates was suggested in previous studies, indicating a possible link of Li/Ca to salinity (*e.g.*, *Marriott et al., 2004*). Our results suggest that a similar relationship may exist for at least some echinoid species.

The absence of strong effects of body size on element ratios suggests that variation in body size variation within and across species is not playing a major role in controlling geochemical signatures. However, specimens analyzed here are primarily adults so ontogenetic effects observed for heavily biomineralized taxa when contrasting juvenile and adult specimens (*e.g.*, *Rosenberg, 1989*) cannot be evaluated in this study.

The micro/nano-scale analyses of the two species (*L. sexiesperforata* and *E. michelini*) indicated the presence of systematic differences in Mg/Ca ratios between different zones within a single test. For both species, the outer imperforate stereom layers that locally form tubercles, were enriched in Mg, and such enrichment may strengthen the outer zone of the test. This was confirmed for both species by nanoindentation analyses indicating that within the same individuals the outer stereom layers exhibited significantly higher nanohardness than the inner stereom microfabrics. These results are consistent with theoretical and empirical studies showing a positive effect of Mg impurities on mechanical properties in biominerals (*e.g.*, *Kunitake et al., 2013*; *Bianco-Stein et al., 2022*; *Gorzelak et al., 2024*). Thus, increased Mg contents in the outer stereom layers may be indicative of active physiological regulation to enhance the mechanical strength of the outermost test parts (tubercles), which undergo constant surface friction and wear. On the other hand, elevated Mg contents are expected to increase skeletal solubility (*e.g.*, *Morse, Andersson & Mackenzie, 2006*), which may be detrimental, especially in the case of spine loss and/or when epidermis is degraded within acidified sediment. However, the higher density of the outer stereom (with much reduced surface-to-volume ratio) relative to that of the underlying stereom appears to mitigate the potential effect of increased skeletal solubility imposed by the elevated Mg content (*e.g.*, *Dery et al., 2014*). The slight offset in Mg/Ca ratios observed between the two specimens of *E. michelini* collected from different environmental settings (Northern Gulf *vs.* Florida Keys) suggests possible influence of salinity on Mg/Ca values (Fig. 7), consistent with the bulk analysis of other specimens (Fig. 4A). However, this offset is minor compared to offsets between the two species suggesting stronger physiological control.

Intriguingly, nearly identical nanohardness values for the two investigated taxa, despite notable differences in the Mg/Ca ratios of their tests suggest that nano-physical properties may be also controlled by factors other than just Mg. The comparable nanohardness of the two specimens of *E. michelini*, which were collected from different salinity regimes and had different Mg/Ca ratios (Fig. 8), further supports the notion that nanohardness

is not exclusively controlled by magnesium. We posit a hypothesis that these species may use different strategies to achieve similar nanohardness. It has been previously demonstrated that nanoindentation hardness can be also affected by nanoscopic organic inclusions, *i.e.,* amino acid content (*Kim et al., 2016*). Indeed, echinoderm biominerals have a nanocomposite structure consisting of tightly aggregated 20–100 nm spherical particles often surrounded by organic macromolecules (*Gorzelak et al., 2013*), and the quantity and quality of this intra-stereomic organic matrix (IOM) may vary by species (*e.g.*, *Ameye et al., 2001*; *Hermans et al., 2011*). Thus, it is possible that *E. michelini* achieved nanohardness comparable to that observed for *L. sexiesperforata* (Figs. 8–10) by modifying its intra-stereomic matrix to compensate for relatively lower magnesium levels. This mechanism could also explain the comparable nanohardness observed for *E. michelini* specimens from different salinity regimes (Figs. 9 and 10). In summary, the nano-scale analyses suggest a substantial physiological control on Mg/Ca ratios across different zones of the test that appear remarkably consistent in nanohardness for the two analyzed species.

The joint consideration of bulk and nano-scale analyses suggests that both physiological and extrinsic factors play a significant role in controlling geochemical signatures of the four analyzed echinoid species, but their relative role varies depending on the element, taxon, and habitat. Geochemical signatures archived in echinoid tests may thus contain both extrinsic environmental and intrinsic biological information that may be difficult to disentangle given our current knowledge.

## ACKNOWLEDGEMENTS

We thank Michał Gloc and Krzysztof Kulikowski (both from Warsaw University of Technology, Faculty of Materials Science and Engineering) for help in performing nanomechanical measurements, and Kamil Humański for help in preparation of bulk samples for geochemical analyses. We thank journal Editor Blanca Figuerola, and three reviewers (Andreas Kroh, Jeff Thompson and one anonymous reviewer) for numerous useful comments and suggestions, which greatly improved our paper.

### Funding

Specimen collection for this project was supported by the National Science Foundation grant EAR-2127623 (University of Florida). The project was funded by NCN grant 2020/37/B/ST10/01460. There was no additional external funding received for this study. The funders had no role in study design, data collection and analysis, decision to publish, or preparation of the manuscript.

### Grant Disclosures

The following grant information was disclosed by the authors:
National Science Foundation: EAR-2127623.
NCN: 2020/37/B/ST10/01460.

## Competing Interests

The authors declare there are no competing interests.

## Author Contributions

- Przemysław Gorzelak conceived and designed the experiments, performed the experiments, analyzed the data, authored or reviewed drafts of the article, and approved the final draft.
- Luis Torres, Jr. analyzed the data, authored or reviewed drafts of the article, and approved the final draft.
- Dorota Kołbuk performed the experiments, analyzed the data, authored or reviewed drafts of the article, and approved the final draft.
- Tobias B. Grun analyzed the data, authored or reviewed drafts of the article, and approved the final draft.
- Michał Kowalewski conceived and designed the experiments, performed the experiments, analyzed the data, prepared figures and/or tables, authored or reviewed drafts of the article, and approved the final draft.

## Field Study Permissions

The following information was supplied relating to field study approvals (i.e., approving body and any reference numbers):

All surveying and collecting activities were carried out within the scope of the collecting permits SAL-18-1294A-SR, SAL-19-2195-SR, SAL-19-2195A-SR, SAL-22-2195-SR, and SAL-22-2195A-SR issued by the Florida Fish and Wildlife Conservation Commission.

## Data Availability

The R script and data are available in the Supplementary Files.

## Supplemental Information

Supplemental information for this article can be found online at http://dx.doi.org/10.7717/peerj.18688#supplemental-information.

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
