# Peer review of "Geochemical signatures and nanomechanical properties of echinoid tests from nearshore habitats of Florida: environmental and physiological controls on echinoid biomineralization"

_PeerJ, doi:10.7717/peerj.18688_

## Round 0.1 · original submission · Major Revisions

Your manuscript has been evaluated by three peer reviewers, and the reviewer comments are appended below.

I am glad to tell you that the reviews are positive. However, they are also advising that your manuscript should be revised at some places before being accepted (e.g. statistical analyses and figures).

Based on the referees' recommendations, I arrive at this decision: The manuscript does merit publication in PeerJ but it is not acceptable in its current form and needs a major revision based on the reviews and my general editorial comments below. I therefore invite you to resubmit a revised version.

Please carefully consider the comments of the reviewers and provide a point-by-point response which clearly defines the changes made.

Thank you for your patience with the evaluation process and for choosing PeerJ.

I look forward to receiving your revised manuscript.

Reviewer 1 ·

Basic reporting

The text is clear, professional English.

The analytical elemental data are given in ratios rather than raw form – and appear to be incomplete with regard to elements detected (although it can’t be conclusively stated from the appendix1.csv table).

The raw data for elemental composition were not provided for the manuscript – only calculated ratios (using calcium in the denominator) were provided, and these appear to be incomplete. Data should be reported in ppm, mole fraction, or whatever form reported by the lab generating the data. In addition, all results should be reported. Even non-detection of certain elements could be significant.

Custom checks:

The Special Activity Licenses from the Florida Fish and Wildlife Conservation Commission are appropriate for the collection of live marine invertebrates within the study areas. Even a standard Saltwater Recreational Fishing License from the FFWCC would be sufficient for the number of specimens taken for the study.

Experimental design

The study is original research within the scope of the journal, at least in the sense that the elemental composition data are new and valuable.

There is, however, a major problem with the analysis. I preface this comment with a quote from F. Chayes, F., 1949, On Ratio Correlation in Petrography, Journal of Geology, Vol. 57, No. 3, pp. 239-254: “The formation of ratios should be confined to those problems in which hypotheses being tested deal with ratios. Absolute measures are always preferable when large numbers of observations must be recorded without benefit of satisfactory hypothesis. Ratios can always be drawn from tables of absolute measures; frequently, absolute measures cannot be reclaimed from tables of ratios.” This is one aspect of what is known as the constant sum problem inherent to compositional data. Standard statistical analyses of compositional data result in spurious correlations. There is an extensive literature on Compositional Data Analysis (CoDA), and a nice R package is available:
(https://cran.r-project.org/web/packages/compositions/index.html).

Validity of the findings

As noted above, the analysis of the data is not statistically robust. Re-analysis of the data using CoDA tools may or may not support the conclusions given in the manuscript.

Additional comments

Title: There doesn’t seem to be anything in the manuscript regarding “geochemical archives”. There is no discussion nor any associated references concerning the preservation of echinoid calcite in the geologic record.

Lines 87-88: Why were these particular elements selected for analysis? Ni is repeated in the list – which is missing Li and Zn, but includes Co. The BVM brochure lists 37 elements for an AQ250 analysis (which doesn’t include Li) and 52 elements for AQ250-EXT (which includes Li). Sulfur, which in general will exist in the -2 state (or as a sulfate ion in the -2 state), is an anion, and rather than replacing a calcium or magnesium cation would be expected to replace a carbonate anion and be controlled by a different kinetic pathway. In that case, the sulfur/carbon ratio would be more significant than the sulfur/calcium ratio.

Line 144: The usage of Encope spp. is not consistent within the text. The only place E. aberrans is mentioned in the text is here (and in the abstract). Elsewhere, E. michelini is always identified.

Line 188: What kind of alcohol?

Line 188: “cleaning” – juvenile scutelline sand dollars of the species studied here have diverticula containing heavy mineral particles – if present, are these completely removed during cleaning?

Line 189: A small figure illustrating what part of the test was sampled would be helpful. This is true for the microprobe samples as well (lines 205-206).

Line 194: Eleven (not 8) elements are listed (the additions are nickel, cobalt, and boron).

Lines 195-196: How does reporting results as ratios over calcium account for organic inclusions? If trace elements are detected from organic inclusions how does this correct for the problem (if it is even a problem)?

The reporting of the data as ratios with calcium in the denominator ignores the fact that the proportion of calcium is negatively correlated to the proportions of all the other elements. Consider a simple example: a 3-component system of Ca, Mg, and some minor element X. Let the Mg/Ca ratio vary between 0 and 1 (i.e., pure calcite to pure dolomite), but let X be constant (see table below). While the actual proportion of X in a sample will be constant across all samples, the X/Ca ratio will vary according to the Mg/Ca ratio. There will be an apparent perfect correlation between X/Ca and Mg/Ca, although the correlation between X and Mg is, in reality, 0.

Sample Ca Mg X Total Mg/Ca X/Ca
1 0.90 0.00 0.10 1 0.000 0.111
2 0.80 0.10 0.10 1 0.125 0.125
3 0.70 0.20 0.10 1 0.286 0.143
4 0.60 0.30 0.10 1 0.500 0.167
5 0.50 0.40 0.10 1 0.800 0.2
6 0.45 0.45 0.10 1 1.000 0.222

Lines 200-201: The BVM brochure lists detection limits (AQ250 and AQ250-EXT) for Li and Pb as 0.1 and 0.01 ppm, respectively.

Line 202 (Nanoindentation section): It is known that the hardness of calcite varies significantly in accordance with crystal axis orientation, but this was not accounted for in the methods discussion. This is where a diagram showing exactly how these samples were cut could be important.

Line 209: What are the units of nanohardness?

Line 288: Why restrict the PCA to only the 8 elemental ratios? Since the data are pre-processed by taking z-scores, why not include water temperature and salinity in the PCA?

Line 383: Should it be Ba/Ca, Pb/Ca, etc.? Also at line 390.

Line 416: Paracentrotus

Figure 4: What do the different symbol sizes represent?

Figures 6 & 8: The Mg/Ca ratios here are in mol/mol (not mmol/mol). The values are correct in Fig. 7.

Appendix1.csv: The units should be included in the table.

Appendix2.csv: The units should be included in the table.

·

Basic reporting

The present manuscript is a valuable contribution to the knowledge of echinoderm geochemistry – a field where relatively little data is available yet. As such, the study is a welcome step forward. The manuscript is well written and previous work is acknowledged sufficiently. Figures, tables, code and raw data are well organized and available (raw data and R code in GitHub repository). Due to the limited sample size and high variability in the dataset conclusions drawn cannot be generalized broadly, but provide the basics for future work.

Regarding the statement in line 134 ("These alternative results...") I would argue that the authors should provide the alternative results/figures in a supplementary file rather than expect the reader to modify an R script and rerun the analyses.

Experimental design

All good, as far as I can tell. I would ask the authors to add inventory numbers for the studied specimens in the supplementary tables though and to clearly mention where these specimens/samples will be deposited permanently.

Regarding sampling for bulk geochemical analyses, it is unclear from the text (Line 189ff. “… approximately 1/5 of each specimen […] was powdered and homogenized …”) if only the coronal plates were used in these analyses or if part of the lantern was included. This is very relevant, since it is well established that the urchin teeth show deviant geochemical signatures compared to the coronal plates (increased Sr and Mg contents – see summary in our review: KROH, A. & NEBELSICK, J.H. (2010): Echinoderms and Oligo-Miocene carbonate systems: potential applications in sedimentology and environmental reconstruction [Chapter 12]. – In: MUTTI, M., PILLER, W.E. & BETZLER, C. (eds): Carbonate Systems During the Olicocene-Miocene Climatic Transition: (Special Publication 42 of the IAS). 1. Auflage – pp. 201–226, New York (John Wiley & Sons).).

It is also unclear if the four specimens used for nano-hardness are a subset of the 43 specimens used in bulk geochemical analyses or four different specimens not used in the geochemical analyses.

Line 191: unclear how many aliquots were generated

Validity of the findings

The authors have provided the raw data and have been careful in their interpretation of the results acknowledging the limited sample size (both in number and taxonomic breadth).
I would, however, recommend to omit statements such as “… may not only allow us to better understand their biomineralization, taphonomy and diagenesis, but also determine the utility of fossil clypeasteroids as geochemical archives” (Line 103ff.) and “… but may eventually allow for insightful analyses of fossil echinoids from the 485 paleo-environmental and eco-evolutionary perspectives” (Line 484ff.). – It has yet to be investigated how resilient the analysed geochemical signatures are in relation to diagenetic changes during taphonomy and fossilization, which in echinoderms almost invariably (with few exceptions) encompasses transformation of the high-Mg skeleton in low-Mg calcite. These statements raise the expectations that results from modern settings using live specimens can be directly transferred to fossil remains, which very likely will only be possible under very special conditions.

Lines 445-449: here the authors relate enhanced Mg-content to enhanced mechanical strength of the outermost stereo layer, yet in lines 370 to 372 they say that the nano-physical properties of the two taxa in discussion were similar despite different Mg-content. These two statements appear to be in conflict – please revisit these parts and rephrase as necessary.

Additional comments

In-text citations: please unify usage of comas, semicolons and blanks – at present sometimes a coma is used btw. author and year and sometimes not, likewise some citations are separated by comas others by semicolons etc. (e.g. lines 52, 388,...)

Units: add blanks between the numeric value and the unit (e.g. line 126: “21 m” instead of “21m”; also in lines 173, 174, 191)

Line 123: add “the” before “lowest”

Line 363: closing bracket should not be in bold font

Line 397: “Fig. 4” should not be in bold font

·

Basic reporting

This manuscript from Gorzelak et al. is a welcome addition to our understanding of echinoderm biomineralization, with implications for biologists, chemists, and palaeontologists. An important, and often less-understood component of our understanding of echinoderm biomineralization is the relative contributions of biological and environmental controls on the biomineralization process. Gorzelak et al. show herein that the impact of the environment on regulating skeletal geochemistry is mixed, with some species exhibiting tighter control on the trace element composition of their skeleton than others. Furthermore, the relative importance of the environment does not effect the incorporation of all trace elements equally, with some (Mg, Sr, Li), more tightly regulated by the environment than others (Ba, Pb, Mn). The manuscript is clear, and well-written, and, by comparison to many other papers on related topics, very statistically rigorous and conservative with its interpretations. Overall, I recommend minor revisions to the manuscript, and have listed the minor changes I suggest below. I additionally congratulate the authors on this very nice contribution.

The background is clear and to-the-point, and references the appropriate literature. However, one thing that needs to be addressed is the taxonomic placement of the studied species (starting on line 88). Recent changes in echinoid phylogeny resulting from phylogenomics (Mongiardino Koch et al. 2021 being the most recent) have shown that the former Clypeasteroids are no longer a monophyletic group. One of the implications of this is that the Mellitids (which includes all non-Clypeaster) taxa in the manuscript discussed herein are no longer clypeasteroids. So, please make the change as required throughout the manuscript. Clypeaster is a clypeasteroid, but the other included taxa are Scutelloids (and not clypeasteroids). I think you get away with this in the methods by referring to the family level, but in the intro it needs to be changed.



Line 60. It would be nice to have some citations to back up the discussed “pitfalls”

Line 104. I think a mention/short discussion of why you chose to carry out the nanohardness analyses (ie what does it bring to the table) is necessary, and is currently lacking.

Experimental design

For the materials and methods, if you have them, I think that the description of the different species could be improved upon/made clearer with some pictures/images of the study species.

Why did you choose to analyse the 8 elements that you did? Some justification of this decision would be a welcome addition to the manuscript. Some are obvious (Sr, Mg), but others less-so.


Why did you only carry out the microhardness analyses on two of the studied taxa?

Validity of the findings

All findings are valid, and rigorously backed up by statistical analysis. The comparison of hardness and Mg/content, and the comparison of hardness across different types of stereom is awesome, and a particularly welcome contribution. All data is available and robust.

Additional comments

It was unclear to me what the size of the symbols on the PCA represented. This should be clarified in the figure caption.

Line 382. Remove “a”.

Line 391. Remove “the”.

---

## Round 0.2 · Minor Revisions

The authors have greatly improved their review according to the referees' comments although some minor revisions are still required (see reviewer comments).

Reviewer 1 ·

Basic reporting

The text is clear, professional, and well-organized.

Experimental design

No comment

Validity of the findings

No comment

Additional comments

The authors have satisfied my primary concern by including a full raw data set. I don’t necessarily agree with all of their responses to my concerns regarding the analysis of the data but accept their justifications for their approach. I feel that the paper can be accepted when the minor items listed below have been addressed.

Detailed comments/corrections

Line 256: The pdf file conversion appears to have mistranslated the formula 0.7*L by not vertically centering the ‘*’ multiplication operator (I note the same problem here - there is apparently a problem in translating MS Word formulas).

References:

Allen et al. 2011 – italicize journal name and italicize Orbulina universa.

Brustolin et al. 2016 – italicize journal name.

Gorzelak et al. 2024 – reformat DOI for consistency.

Gray et al. 2023 – reformat DOI for consistency.

Henehan et al. 2015 - italicize journal name, italicize Globigerinoides ruber and reformat DOI for consistency.

Knapp et al. 2012 - italicize journal name and reformat DOI for consistency.

Kucera & Malmgren 1998 - italicize journal name and add DOI (available from Crossref).

Marriott et al. 2004 - italicize journal name and add DOI (available from Crossref).

Mooi & Peterson 2000 - reformat DOI for consistency.

Richter & Bruckschen 1998 – italicize Echinocyamus pusillus.

Telford et al 1987 – italicize Clypeaster.

Ulrich et al. 2021 Title should be in sentence case, italicize journal name, doi in uppercase.

Wit et al 2013 – italicize journal name, add DOI (available from Crossref). Add a carriage return before Weber 1969.

Weber 1969 - add DOI (available from Crossref).

Figures and Tables:

Figure 1. The different colors of symbols are not clearly explained in the caption or a legend. If nothing else, the gray shading between the state and the symbols should be different.

Figure 2. The figure caption describes black dots and gray bars – but not the other colors (which apparently are meant to match the symbols in Figure 1).

Figure 3. Nice figure, but maybe lightly shade the regions between the red lines to distinguish from the simple linear sections used for nanohardness

Figure 4. These graphs are difficult to read. Because the caption indicates that the focus is the variation of element ratio by environment (species plays only a minor role), it would be clearer to give each species a different symbol and only use 3 colors for environment (as you do in the other figures).

Figure 5. I only note that you use 0.7*L in the text (Line 256 - also see the comment above for that line)

Tables 4 and 5. It is obvious that the Mg/Ca ratio is mmol/mol, but it needs to be explicitly stated that this is the case for all the listed ratios (i.e., that a factor of 1000 has been applied to the unitless ratios).

Supplemental Data:

The Appendix numbering in the header lines of the supplemental files is missing Appendix 6, and S3 and S4 are both numbered Appendix 2. Since the term ‘Appendix’ is not referenced in the text, perhaps this should be removed from all the supplemental table headers. I have not checked how this may affect the read commands in the R code when someone has to figure out which file is which.

Supplemental Data Table S2: The specimen ID numbers differ in format between this table and tables S3 and S4 – specifically, S2 and S4 use periods and S3 uses commas.

Supplemental Data Table S3: The ‘<’ values all use European decimal notation while the other values use English format – they should probably all use the English format. Note comment for table S2 about specimen ids.

Supplemental Data Table S4: Note comment for table S2 about specimen ids.

---

## Round 0.3 · accepted · Accept

The authors have addressed all of the reviewers' comments and I am pleased to inform that the paper has been accepted for publication without further changes.

Reviewer 1 ·

Basic reporting

No comment

Experimental design

No comment

Validity of the findings

No comment

Additional comments

All the changes are satisfactory and I think the manuscript can be accepted as is.

My comment about the 0.7*L was simply to note that the asterisk, which would normally be vertically centered to represent a multiplication operator, has been shifted to a superscript position when the original formula (I assume) was translated into a pdf. That is, this is strictly a formatting error almost certainly resulting from the PeerJ software. Replacing the * with a x seems to have resolved the problem.